# Uncertainty-modulated prediction errors in cortical microcircuits

**Katharina Anna Wilmes\*, Mihai A Petrovici, Shankar Sachidhanandam, Walter Senn**

Department of Physiology, University of Bern, Bern, Switzerland

### eLife Assessment

This **important** study introduces a new cortical circuit model for predictive processing. Simulations effectively illustrate that, with appropriate synaptic plasticity, a canonical layer 2/3 cortical circuit - comprising two classes of interneurons providing subtractive and divisive inhibition - can generate uncertainty-modulated prediction errors by pyramidal neurons. The model is **compelling**; although it relies on many assumptions and has not yet been compared directly to data, the model does align with empirical observations and yields a range of testable predictions. The study is expected to be of great interest to those involved in cortical and predictive processing research.

**\*For correspondence:**
katharina.wilmes@unibe.ch

**Competing interest:** The authors declare that no competing interests exist.

**Abstract** Understanding the variability of the environment is essential to function in everyday life. The brain must hence take uncertainty into account when updating its internal model of the world. The basis for updating the model are prediction errors that arise from a difference between the current model and new sensory experiences. Although prediction error neurons have been identified in layer 2/3 of diverse brain areas, how uncertainty modulates these errors and hence learning is, however, unclear. Here, we use a normative approach to derive how uncertainty should modulate prediction errors and postulate that layer 2/3 neurons represent uncertainty-modulated prediction errors (UPE). We further hypothesise that the layer 2/3 circuit calculates the UPE through the subtractive and divisive inhibition by different inhibitory cell types. By implementing the calculation of UPEs in a microcircuit model, we show that different cell types can compute the means and variances of the stimulus distribution. With local activity-dependent plasticity rules, these computations can be learned context-dependently, and allow the prediction of upcoming stimuli and their distribution. Finally, the mechanism enables an organism to optimise its learning strategy via adaptive learning rates.

## Introduction

Decades of cognitive research indicate that our brain maintains a model of the world, based on which it can make predictions about upcoming stimuli (*Kveraga et al., 2007*; *Cohen et al., 2011*). Predicting the sensory experience is useful for both perception and learning: Perception becomes more tolerant to uncertainty and noise when sensory information and predictions are integrated (*Payzan-LeNestour and Bossaerts, 2011*). Learning can happen when predictions are compared to sensory information, as the resulting prediction error indicates how to improve the internal model. In both cases, the uncertainties (associated with both the sensory information and the internal model) should determine how much weight we give to the sensory information relative to the predictions, according to theoretical accounts. Behavioural and electrophysiological studies indicate that humans indeed estimate uncertainty and adjust their behaviour accordingly (*Payzan-LeNestour and Bossaerts, 2011*; *Walker et al., 2019*; *Goris et al., 2018*; *Cannon et al., 2021*; *Körding and Wolpert, 2004*). The neural mechanisms

underlying uncertainty and prediction error computation are, however, less well understood. Recently, the activity of individual neurons of layer 2/3 cortical circuits in diverse cortical areas of mouse brains has been linked to prediction errors (visual, *Keller et al., 2012*; *Zmarz and Keller, 2016*; *Fiser et al., 2016*; *Attinger et al., 2017*; *Gillon et al., 2021*), auditory (*Eliades and Wang, 2008*; *Keller and Hahnloser, 2009*), somatosensory (*Ayaz et al., 2019*), and posterior parietal (*Raltschev et al., 2023*). Importantly, prediction errors could be associated with learning (*Jordan and Keller, 2023*). Prediction error neurons are embedded in neural circuits that consist of heterogeneous cell types, most of which are inhibitory. It has been suggested that prediction error activity results from an imbalance of excitatory and inhibitory inputs (*Hertäg and Sprekeler, 2020*; *Hertäg and Clopath, 2022*), and that the prediction is subtracted from the sensory input [see e.g. *Rao and Ballard, 1999*; *Attinger et al., 2017*], possibly mediated by so-called somatostatin-positive interneurons (SSTs) (*Attinger et al., 2017*). How uncertainty is influencing these computations has not yet been investigated. Prediction error neurons receive inputs from a diversity of inhibitory cell types, the role of which is not completely understood. Here, we hypothesise that one role of inhibition is to modulate the prediction error neuron activity by uncertainty.

In this study, we use both analytical calculations and numerical simulations of rate-based circuit models with different inhibitory cell types to study circuit mechanisms leading to uncertainty-modulated prediction errors. First, we derive that uncertainty should divisively modulate prediction error activity and introduce uncertainty-modulated prediction errors (UPEs). We hypothesise that, first, layer 2/3 prediction error neurons reflect such UPEs. Second, we hypothesise that different inhibitory cell types are involved in calculating the difference between predictions and stimuli, and in the uncertainty modulation, respectively. Based on experimental findings, we suggest that SSTs and PVs play the respective roles. We then derive biologically plausible plasticity rules that enable those cell types to learn the means and variances from their inputs. Notably, because the information about the stimulus distribution is stored in the connectivity, single inhibitory cells encode the means and variances of their inputs in a context-dependent manner. Layer 2/3 pyramidal cells in this model hence encode uncertainty-modulated prediction errors context-dependently. We show that error neurons can additionally implement out-of-distribution detection by amplifying large errors and reducing small errors with a nonlinear fI-curve (activation function). Finally, we demonstrate that UPEs effectively mediate an adjustable learning rate, which allows fast learning in high-certainty contexts and reduces the learning rate, thus suppressing fluctuations in uncertain contexts.

## Results
### Normative theories suggest uncertainty-modulated prediction errors (UPEs)

In a complex, uncertain, and hence partly unpredictable world, it is impossible to avoid prediction errors. Some prediction errors will be the result of this variability or noise, other prediction errors will be the result of a change in the environment or new information. Ideally, only the latter should be used for learning, that is updating the current model of the world. The challenge our brain faces is to learn from prediction errors that result from new information, and less from prediction errors that result from noise. Hence, intuitively, if we learned that a kind of stimulus or context is very variable (high uncertainty), then a prediction error should have only little influence on our model. Consider a situation in which a person waits for a bus to arrive. If they learned that the bus is not reliable, another late arrival of the bus does not surprise them and does not change their model of the bus (*Figure 1A*). If, on the contrary, they learned that the kind of stimulus or context is not very variable (low uncertainty), a prediction error should have a larger impact on their model. For example, if they learned that buses are reliable, they will notice that the bus is late and may use this information to update their model of the bus (*Figure 1A*). This intuition of modulating prediction errors by the uncertainty associated with the stimulus or context is supported by both behavioural studies and normative theories of learning. Here we take the view that uncertainty is computed and represented on each level of the cortical hierarchy, from early sensory areas to higher level brain areas, as opposed to a task-specific uncertainty estimate at the level of decision-making in higher level brain areas (*Figure 1B*) [see this review for a comparison of these two accounts: *Koblinger et al., 2021*].

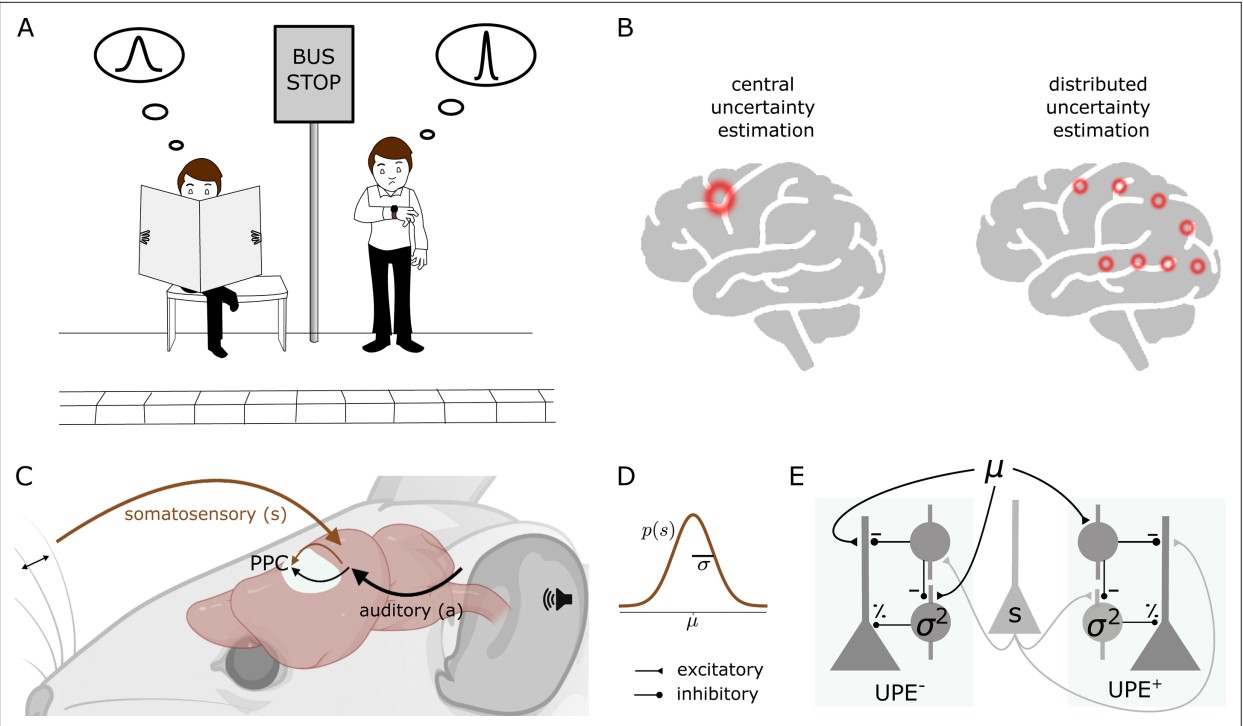

**Figure 1.** Distributed uncertainty-modulated prediction error computation in cortical circuits. (**A**) A person who learned that buses are unreliable has a prior expectation, which can be described by a wide Gaussian distribution of expected bus arrival times. When the bus does not arrive at the scheduled time, this person is not surprised and remains calm, as everything happens according to their model of the world. On the other hand, a person who learned that buses are punctual, which can be described by a narrow distribution of arrival times, may notice that the bus is late and get nervous, as they expected the bus to be punctual. This person can learn from this experience. If they always took this particular bus, and their uncertainty estimate is accurate, the prediction error could indicate that the bus schedule changed. (**B**) Models of uncertainty representation in cortex. Some models suggest that uncertainty is only represented in higher level areas concerned with decision-making (left). In contrast, we propose that uncertainty is represented at each level of the cortical hierarchy (right, shown is the visual hierarchy as an example). (**C**) A mouse learns the association between a sound (*a*) and a whisker deflection (*s*). The posterior parietal cortex (PPC) receives inputs from both somatosensory and auditory cortex. (**D**) The whisker stimulus intensities are drawn from a Gaussian distribution with mean μ and standard deviation $\sigma$. (**E**) Negative (left) and positive (right) prediction error circuit consisting of three cell types: layer 2/3 pyramidal cells (triangle), somatostatin-positive interneurons (SST, circle) and parvalbumin-positive interneurons (PV). SSTs represent the mean prediction in the postive circuit and the stimulus in the negative circuit, and PVs represent the variance.

Before we suggest how cortical circuits compute such uncertainty-modulated prediction errors, we consider the normative solution to a simple association that a mouse can learn. The setting we consider is to predict a somatosensory stimulus based on an auditory stimulus. The auditory stimulus *a* is fixed, and the subsequent somatosensory stimulus *s* is variable and sampled from a Gaussian distribution ($s \sim \mathcal{N}(\mu, \sigma)$, **Figure 1C and D**). The optimal (maximum-likelihood) prediction is given by the mean of the stimulus distribution. Framed as an optimisation problem, the goal is to adapt the internal model of the mean $\hat{\mu}$ such that the probability of observing samples *s* from the true distribution of whisker deflections is maximised given this model.

Hence, stochastic gradient ascent learning on the log-likelihood suggests that with each observation *s*, the prediction (corresponding to the internal model of the mean) should be updated as follows to approach the maximum likelihood solution:

$$\Delta \hat{\mu} \propto \frac{\partial}{\partial \hat{\mu}} (\log \mathcal{L}) = \frac{1}{\sigma^2} (s - \hat{\mu}), \tag{1}$$

where L is the likelihood. According to this formulation, the update for the internal model should be the prediction error scaled inversely by the variance $\sigma^2$. Therefore, we propose that prediction errors should be modulated by uncertainty.

## Computation of UPEs in cortical microcircuits

How can cortical microcircuits achieve uncertainty modulation? Prediction errors can be positive or negative, but neuronal firing rates are always positive. Because baseline firing rates are low in layer 2/3 pyramidal cells [e.g., (*Niell and Stryker, 2008*)], positive and negative prediction errors were suggested to be represented by distinct neuronal populations (*Keller and Mrsic-Flogel, 2018*; *Rao and Ballard, 1999*), which is in line with experimental data (*Jordan and Keller, 2020*). We, therefore, decompose the UPE into a positive UPE⁺ and a negative UPE⁻ component (*Figure 1C and D*):

$$\mathrm{UPE} = \mathrm{UPE}^+ - \mathrm{UPE}^- = \frac{1}{\sigma^2}\lfloor s - \mu \rfloor^+ - \frac{1}{\sigma^2}\lfloor \mu - s \rfloor^+, \tag{2}$$

where $\lfloor ... \rfloor^+$ denotes rectification at 0.

It has been suggested that error neurons compute prediction errors by subtracting the prediction from the stimulus input (or vice versa) (*Attinger et al., 2017*). The stimulus input can come from local stimulus-encoding layer 2/3 cells (*Raltschev et al., 2023*). Inhibitory interneurons provide the subtraction, resulting in an excitation-inhibition balance when they match (*Hertäg and Sprekeler, 2020*). To represent a UPE, error neurons need additionally be divisively modulated by the uncertainty. Depending on synaptic properties, such as reversal potentials, inhibitory neurons can have subtractive or divisive influences on their postsynaptic targets. Therefore, we propose that an inhibitory cell type that divisively modulates prediction error activity represents the uncertainty. We hypothesise, first, that in positive prediction error circuits, inhibitory interneurons with subtractive inhibitory effects represent the mean μ of the prediction. They probably either inherit the mean prediction or calculate it locally. Second, we hypothesise that inhibitory interneurons with divisive inhibitory effects represent the uncertainty $\sigma^2$ of the prediction (*Figure 1C and D*), which they calculate locally. A layer 2/3 pyramidal cell that receives these sources of inhibition then reflects the uncertainty-modulated prediction error (*Figure 1E*).

More specifically, we propose, first, that the SSTs are involved in the computation of the difference between predictions and stimuli, as suggested before (*Attinger et al., 2017*). This subtraction could happen on the apical dendrite. Second, we propose that the PVs provide the uncertainty modulation. In line with this, prediction error neurons in layer 2/3 receive subtractive inhibition from somatostatin (SST) and divisive inhibition from parvalbumin (PV) interneurons (*Wilson et al., 2012*). However, SSTs can also have divisive effects, and PVs can have subtractive effects, dependent on circuit and postsynaptic properties (*Seybold et al., 2015*; *Lee et al., 2012*; *Dorsett et al., 2021*).

In the following, we investigate circuits of prediction error neurons and different inhibitory cell types. We start by investigating in local positive and negative prediction error circuits whether the inhibitory cell types can locally learn to predict means and variances, before combining both subcircuits into a recurrent circuit consisting of both positive and negative prediction error neurons.

## Local inhibitory cells learn to represent the mean and the variance given an associative cue

As discussed above, how much an individual sensory input contributes to updating the internal model should depend on the uncertainty associated with the sensory stimulus in its current context. Uncertainty estimation requires multiple stimulus samples. Therefore, our brain needs to have a context-dependent mechanism to estimate uncertainty from multiple past instances of the sensory input. Let us consider the simple example from above, in which a sound stimulus represents a context with a particular amount of uncertainty. In terms of neural activity, the context could be encoded in a higher level representation of the sound. Here, we investigate whether the context representation can elicit activity in the PVs that reflects the expected uncertainty of the situation. To investigate whether the context provided by the sound can cause activity in SSTs and PVs that reflects the mean and the variance of the whisker stimulus distribution, respectively, we simulated a rate-based circuit model consisting of pyramidal cells and the relevant inhibitory cell types. This circuit receives both the sound and the whisker stimuli as inputs.

## SSTs learn to estimate the mean

With our circuit model, we first investigate whether SSTs can learn to represent the mean of the stimulus distribution. In this model, SSTs receive whisker stimulus inputs $s$, drawn from Gaussian

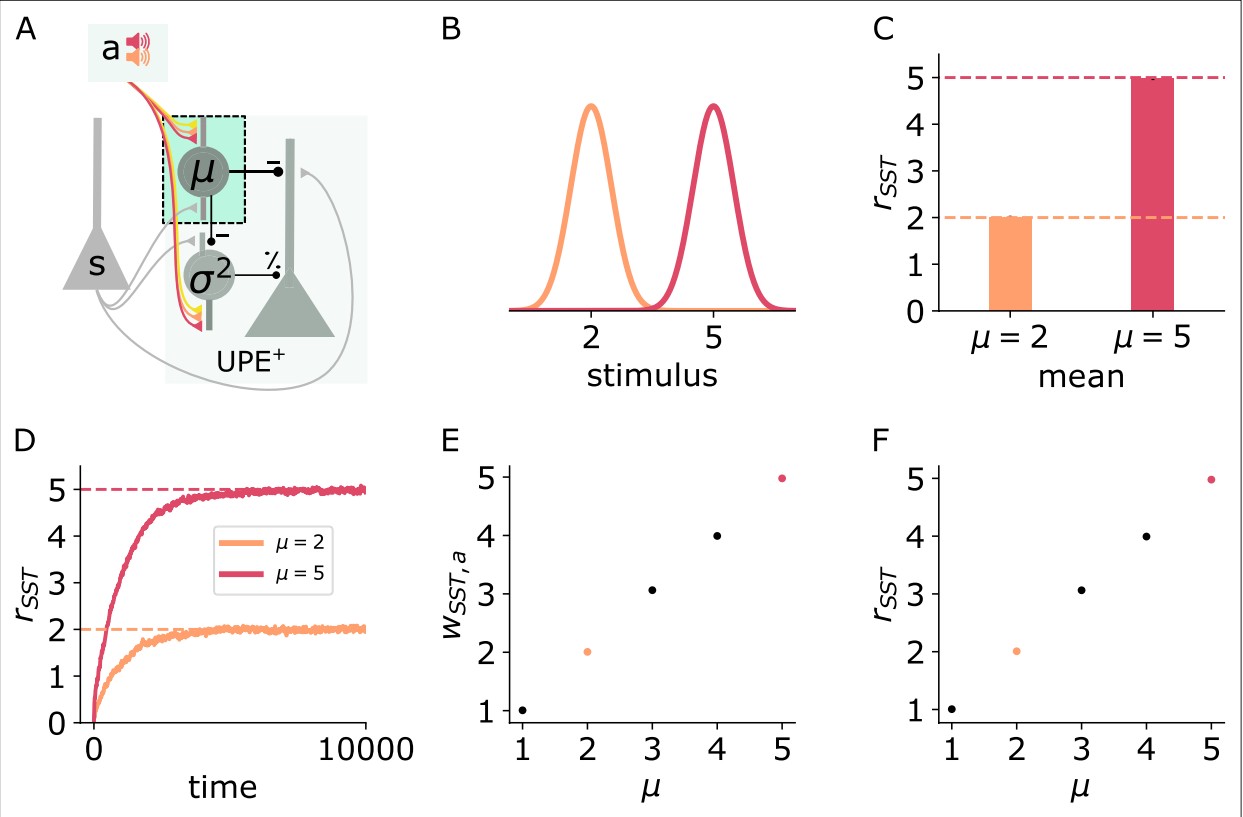

**Figure 2.** SSTs learn to represent the mean context-dependently. Illustration of the changes in the positive prediction error circuit. Thicker lines denote stronger weights. (**B**) Two different tones (red, orange) are associated with two somatosensory stimulus distributions with different means (red: high, orange: low). (**C**) SST firing rates (mean and std) during stimulus input. (**D**) SST firing rates over time for low (orange) and high (red) stimulus means. (**E**) Weights (mean and std) from sound *a* to SST for different values of μ. (**F**) SST firing rates (mean and std) for different values of μ. Mean and std were computed over 1000 data points from timestep 9000–10,000.

distributions (*Figure 2B*), and an input from a higher level representation of the sound *a* (which is either on or off, see *Equation 9* in Methods). The connection weight from the sound representation to the SSTs is plastic according to a local activity-dependent plasticity rule (see *Equation 10*). The aim of this rule is to minimise the difference between the activation of the SSTs caused by the sound input (which has to be learned) and the activation of the SSTs by the whisker stimulus (which nudges the SST activity in the right direction). The learning rule ensures that the auditory input itself causes SSTs to fire at the desired rate. After learning, the weight and the average SST firing rate reflect the mean of the presented whisker stimulus intensities (*Figure 2C–F*).

## PVs learn to estimate the variance context-dependently

We next addressed whether PVs can estimate and learn the variance locally. To estimate the variance of the whisker deflections $s$, the PVs have to estimate $\sigma^2[s] = \mathbb{E}_s[(s - \mathbb{E}[s])^2] = \mathbb{E}_s[(s - \mu)^2]$. To do so, they need to have access to both the whisker stimulus $s$ and the mean μ. PVs in PPC respond to sensory inputs in diverse cortical areas [S1:*Sachidhanandam et al., 2016*] and are inhibited by SSTs in layer 2/3, which we assumed to represent the mean. Finally, for calculating the variance, these inputs need to be squared. PVs were shown to integrate their inputs supralinearly (*Cornford et al., 2019*), which could help PVs to approximately estimate the variance.

In our circuit model, we next tested whether the PVs can learn to represent the variance of an upcoming whisker stimulus based on a context provided by an auditory input (*Figure 3A*). Two different auditory inputs (*Figure 3B* purple, green) are paired with two whisker stimulus distributions that differ in their variances (green: low, purple: high). PVs receive both the stimulus input as well as the inhibition from the SSTs, which subtracts the prediction of the mean (*Equation 11*). The synaptic connection from the auditory input to the PVs is plastic according to the same local activity-dependent

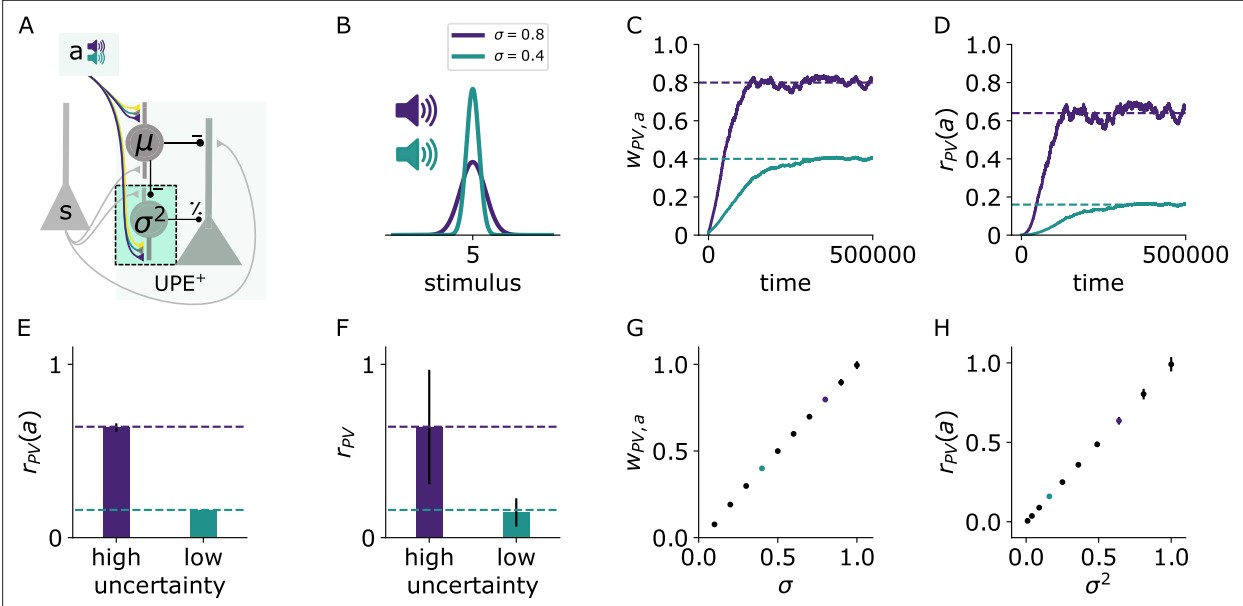

**Figure 3.** PVs learn to estimate the variance context-dependently. (**A**) Illustration of the changes in the positive prediction error circuit. Thicker lines denote stronger weights. (**B**) Two different tones (purple, green) are associated with two somatosensory stimulus distributions with different variances (purple: high, green: low). (**C**) Weights from sound $a$ to PV over time for two different values of stimulus variance (high: $\sigma = 0.8$ [purple], low: $\sigma = 0.4$ [green]). (**D**) PV firing rates over time given sound input (without whisker stimulus input) for low (green) and high (purple) stimulus variance. (**E**) PV firing rates (mean and std) given sound input and whisker stimuli for low and high stimulus variance. (**F**) PV firing rates (mean and std) during sound and stimulus input. (**G**) Weights (mean and std) from sound $a$ to PV for different values of $\sigma$. (**H**) PV firing rates (mean and std) given sound input for different values of $\sigma^2$. Mean and std were computed from 150,000 data points from timestep 450,000–600,000.

The online version of this article includes the following figure supplement(s) for figure 3:

**Figure supplement 1.** Different choice of supralinear activation function for PV.

**Figure supplement 2.** Plastic weights from SST to PV learn to match weights from $s$ to PV.

**Figure supplement 3.** PVs learn to represent the variance given an associative cue in the negative prediction error circuit.

plasticity rule as the connection to the SSTs (*Equation 13*). With this learning rule, the weight onto the PV becomes proportional to $\sigma$ (*Figure 3C*), such that the PV firing rate becomes proportional to $\sigma^2$ on average (*Figure 3D*). The average PV firing rate is exactly proportional to $\sigma^2$ assuming a quadratic activation function $\phi_{PV}(x)$ (*Figure 3D–F and H*) and monotonically increasing with $\sigma^2$ with other choices of activation functions (*Figure 3—figure supplement 1*), both when the sound input is presented alone (*Figure 3D, E and H*) or when paired with whisker stimulation (*Figure 3F*). Notably, a single PV neuron is sufficient for encoding variances of different contexts because the context-dependent $\sigma$ is stored in the connection weights.

To estimate the variance, the mean needs to be subtracted from the stimulus samples. A faithful mean subtraction is only ensured if the weights from the SSTs to the PVs ($w_{PV,SST}$) match the weights from the stimuli $s$ to the PVs ($w_{PV,s}$). The weight $w_{PV,SST}$ can be learned to match the weight $w_{PV,s}$ with a local activity-dependent plasticity rule (see *Figure 3—figure supplement 2* and Appendix).

The PVs can similarly estimate the uncertainty in negative prediction error circuits (*Figure 3—figure supplement 3*). In these circuits, SSTs represent the current sensory stimulus, and the mean prediction is an excitatory input to both negative prediction error neurons and PVs.

## Calculation of the UPE in layer 2/3 error neurons

Embedded in a circuit with subtractive and divisive interneuron types, layer 2/3 pyramidal cells could first compute the difference between the prediction and the stimulus in their dendrites, before their firing rate is divisively modulated by inhibition close to their soma. Layer 2/3 pyramidal cell dendrites can generate NMDA and calcium spikes, which cause a nonlinear integration of inputs. Hence, we took this into account and modelled the integration of dendritic activity as $I_{\text{dend}} = \lfloor w_{\text{UPE},s}\, s - w_{\text{UPE,SST}}\, r_{\text{SST}} \rfloor^k$ with $k$ determining the non-linearity. The total activity of prediction error neurons was modelled by

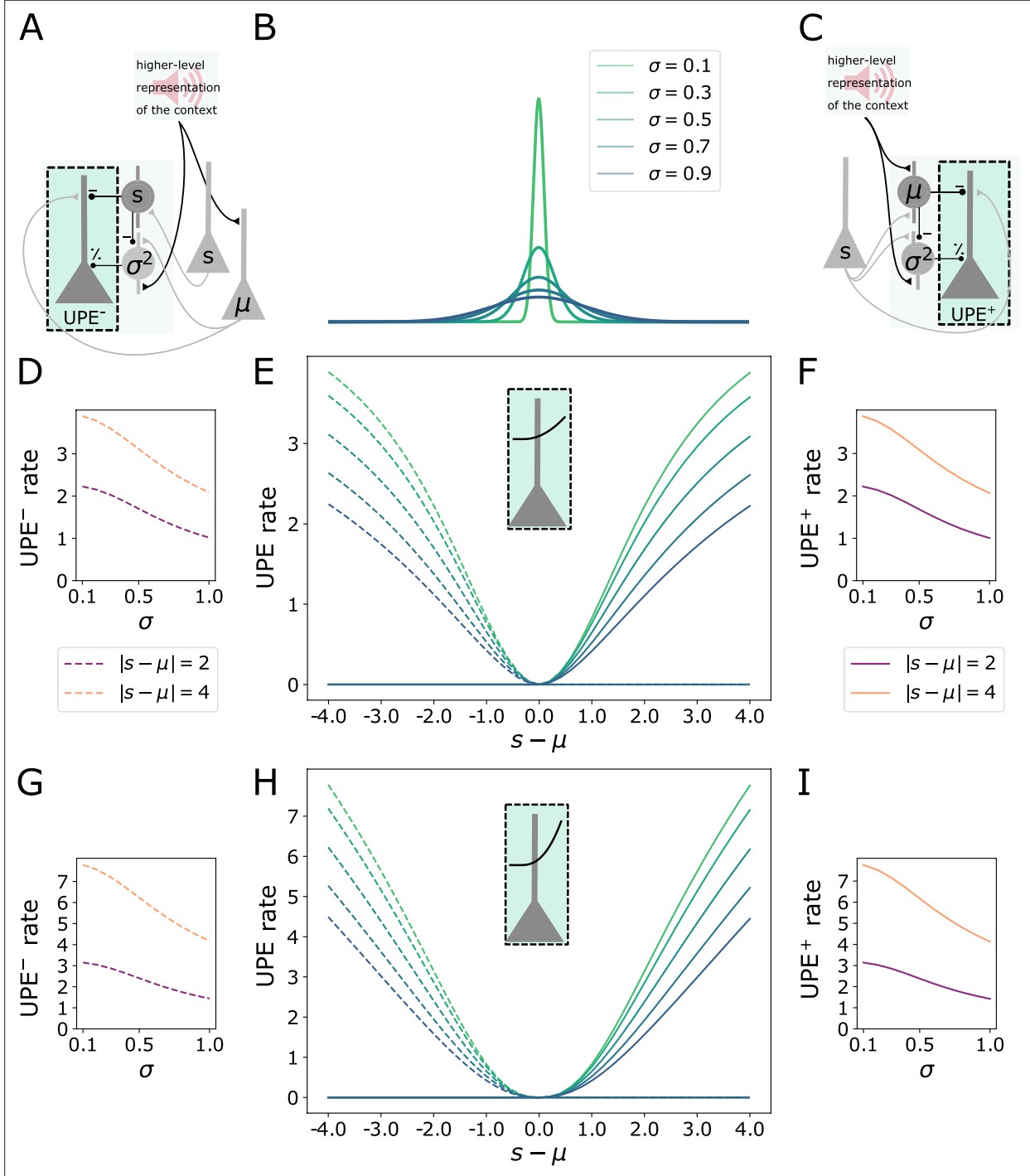

**Figure 4.** Calculation of the UPE in layer 2/3 error neurons. (**A**) Illustration of the negative prediction error circuit. (**B**) Distributions with different standard deviations $\sigma$. (**C**) Illustration of the positive prediction error circuit. (**D**) Firing rate of the error neuron in the negative prediction error circuit (UPE) as a function of $\sigma$ for two values of $|s - \mu|$ after learning μ and $\sigma$. (**E**) Rates of both UPE$^+$ and UPE$^-$-representing error neurons with a nonlinear activation function, where $k = 2.0$, as a function of the difference between the stimulus and the mean ($s - \mu$). (**F**) Firing rate of the error neuron in the positive prediction error circuit (UPE$^+$) as a function of $\sigma$ for two values of $|s - \mu|$ after learning μ and $\sigma$. (**G-I**) same as (**D-F**) for error neurons with $k = 2.5$ with the same legend as in (**D-F**).

The online version of this article includes the following figure supplement(s) for figure 4:

**Figure supplement 1.** Learning the weights from the SSTs to the UPE neurons.

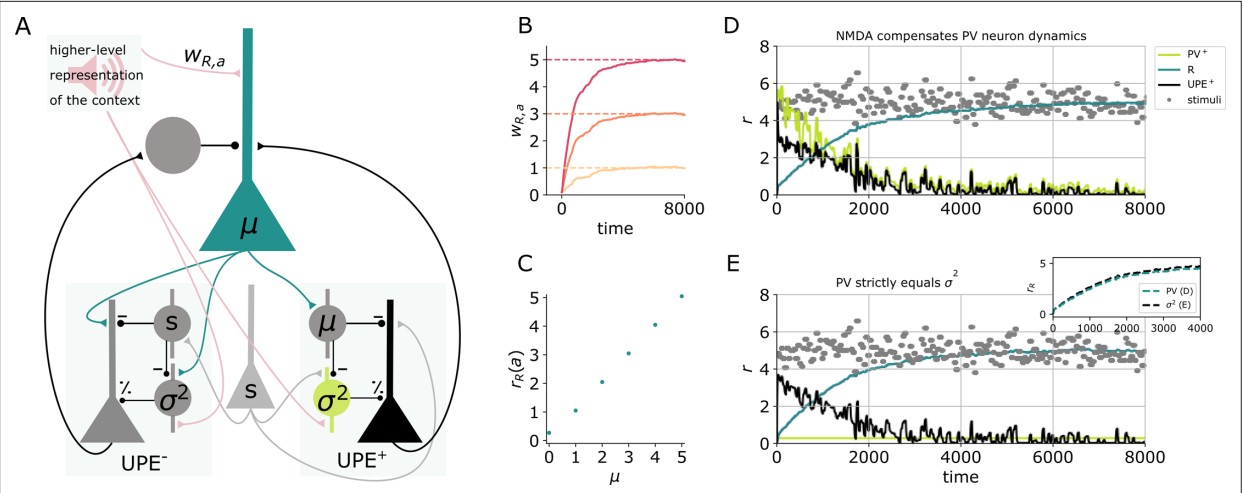

**Figure 5.** Learning the mean representation with UPEs. (**A**) Illustration of the circuit. A representation neuron (turquoise) receives input from both positive and negative prediction error circuits (UPE$^+$ and UPE$^-$) and projects back to them. The UPE$^-$ has a negative impact on the firing rate of the representation neuron ($r_R$). A weight $w_{R,a}$ from the higher level representation of the context given by sound $a$ is learned. (**B**) Weights $w_{R,a}$ over time for different values of μ ($\mu \in [1, 3, 5]$). (**C**) R firing rates given sound input for different values of μ (mean and std over 50,000 data points from timestep 50,000–100,000,, the end of the simulation). (**D**) Activity of the different cell types (PV: light green, R: turquoise, UPE: black) and whisker stimulus samples (grey dots) over time. Learning the mean representation with PVs (light green) reflecting the MSE at the beginning, which is compensated by nonlinear activation of L2/3 neurons (black). The evolution of the mean rate of neuron $R$ (turquoise) is similar to the perfect case in (**E**). (**E**) Same colour code as in (**D**). Inset shows comparison to (**D**) Learning the mean representation assuming PVs (light green) perfectly represent the variance.

The online version of this article includes the following figure supplement(s) for figure 5:

**Figure supplement 1.** PV firing rates are proportional to the variance in the recurrent circuit model.

$$\tau_E \frac{dr_{\text{UPE}}}{dt} = -r_{\text{UPE}} + \phi \left( \frac{I_{\text{dend}}}{I_0 + w_{\text{UPE,PV}} \, r_{\text{PV}}} \right). \tag{3}$$

The nonlinear integration of inputs is beneficial when the mean input changes and the current prediction differs strongly from the new mean of the stimulus distribution.

For example, if the mean input increases strongly, the PV firing rate will increase for larger errors and inhibit error neurons more strongly than indicated by the learned variance estimate. The pyramidal nonlinearity compensates for this increased inhibition by PVs, such that in the end, layer 2/3 cell activity reflects an uncertainty-modulated prediction error (**Figure 4D–F**) in both negative (**Figure 4A**) and positive (**Figure 4C**) prediction error circuits. A stronger nonlinearity (**Figure 4G–I**) has the effect that error neurons elicit larger responses to larger prediction errors.

To ensure a comparison between the stimulus and the prediction, the inhibition from the SSTs needs to match the excitation, which it is compared to, in the UPE neurons: In the positive PE circuit, the weights from the SSTs representing the prediction to the UPE neurons need to match the weights from the stimulus $s$ to the UPE neurons. In the negative PE circuit, the weights from SSTs representing the stimulus to the negative UPE neurons need to match the weights from the mean representation to the UPE neurons, respectively. In line with previous proposals, error neuron activity signals the breaking of EI balance (**Hertäg and Sprekeler, 2020**; **Barry and Gerstner, 2024**). With inhibitory plasticity (target-based, see Appendix), the weights from the SSTs can learn to match the incoming excitatory weights (**Figure 4—figure supplement 1**).

## Interactions between representation neurons and error neurons

The theoretical framework of predictive processing includes both prediction error neurons and representation neurons, the activity of which reflects the internal representation and should hence be compared to the sensory information. To make predictions for the activity of representation neurons, we expand our circuit model with this additional cell type. We first show that a representation neuron R can learn a representation of the stimulus mean given inputs from L2/3 error neurons. The representation neuron receives inputs from positive and negative prediction error neurons and from a higher

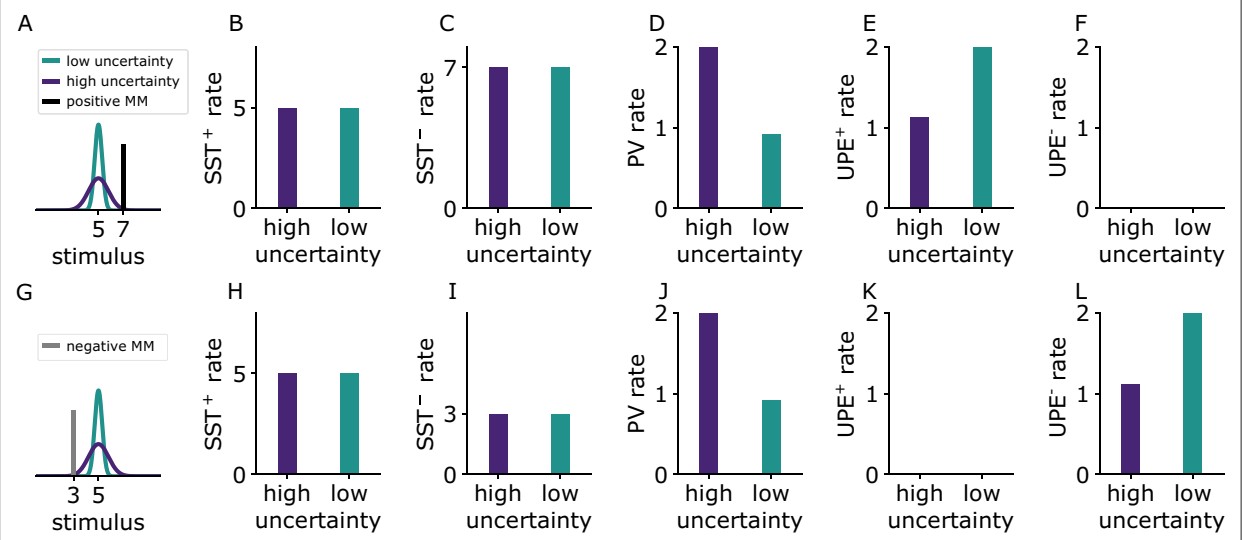

**Figure 6.** Cell-type-specific experimentally testable predictions. (**A**) Illustration of the two experienced stimulus distributions with different variances that are associated with two different sounds (green, purple). The presented mismatch (MM) stimulus (black) is larger than expected (positive prediction error). (**B-F**) Simulated firing rates of different cell types to positive prediction errors when a sound associated with high (purple) or low (green) uncertainty is presented. (**G**) As in (**A**) the presented mismatch stimulus (grey) is smaller than expected (negative prediction error). (**H-L**) Firing rates of different cell types to the negative mismatch when a sound associated with high (purple) or low (green) uncertainty is presented. Because the firing rate predictions are equal for PV⁺ and PV⁻, we only show the results for PV⁺ in the figure.

level representation of the sound $a$ (*Figure 5A*). It sends its current mean estimate to the error circuits by either targeting the SSTs (in the positive circuit) or the pyramidal cells directly (in the negative circuit). Hence in this recurrent circuit, the SSTs inherit the mean representation instead of learning it. After learning, the weights $w_{R,a}$ from the sound representation to the representation neuron and the average firing rate of this representation neuron reflects the mean of the stimulus distribution (*Figure 5B and C*).

As discussed earlier, pyramidal cells tend to integrate their dendritic inputs nonlinearly due to NMDA spikes. We here show that a circuit with prediction error neurons with a dendritic nonlinearity (as in *Figure 4*) approximates an idealised circuit, in which the PV rate perfectly represents the variance (*Figure 5D and E*, see inset for comparison of the two models). The dendritic nonlinearity can hence compensate for PV neuron dynamics. Also in this recurrent circuit, PVs learn to reflect the variance, as the weight from the sound representation $a$ is learned to be proportional to $\sigma$ (*Figure 5— figure supplement 1*).

## Predictions for different cell types

Our model makes predictions for the activity of different cell types for positive and negative prediction errors (e.g. when a mouse receives whisker stimuli that are larger (*Figure 6A*, black) or smaller (*Figure 6G*, grey) than expected) in contexts associated with different amounts of uncertainty (e.g. the high-uncertainty (purple) versus the low-uncertainty (green) context are associated with different sounds). Our model suggests that there are two types of interneurons that provide subtractive inhibition to the prediction error neurons (presumably SST subtypes): in the positive prediction error circuit (SST⁺), they signal the expected value of the whisker stimulus intensity (*Figure 6B and H*). In the negative prediction error circuit (SST⁻) they signal the whisker stimulus intensity (*Figure 6C, I*). We further predict that interneurons that divisively modulate prediction error neuron activity represent the uncertainty (presumably PVs). Those do not differ in their activity between positive and negative circuits and may even be shared across the two circuits: in both positive and negative prediction error circuits, these cells signal the variance (*Figure 6D and J*). L2/3 pyramidal cells that encode prediction errors signal uncertainty-modulated positive prediction errors (*Figure 6E*) and uncertainty-modulated negative prediction errors (*Figure 6L*), respectively. Finally, the existence of so-called internal representation neurons has been proposed (*Keller and Mrsic-Flogel, 2018*). In our case, those neurons

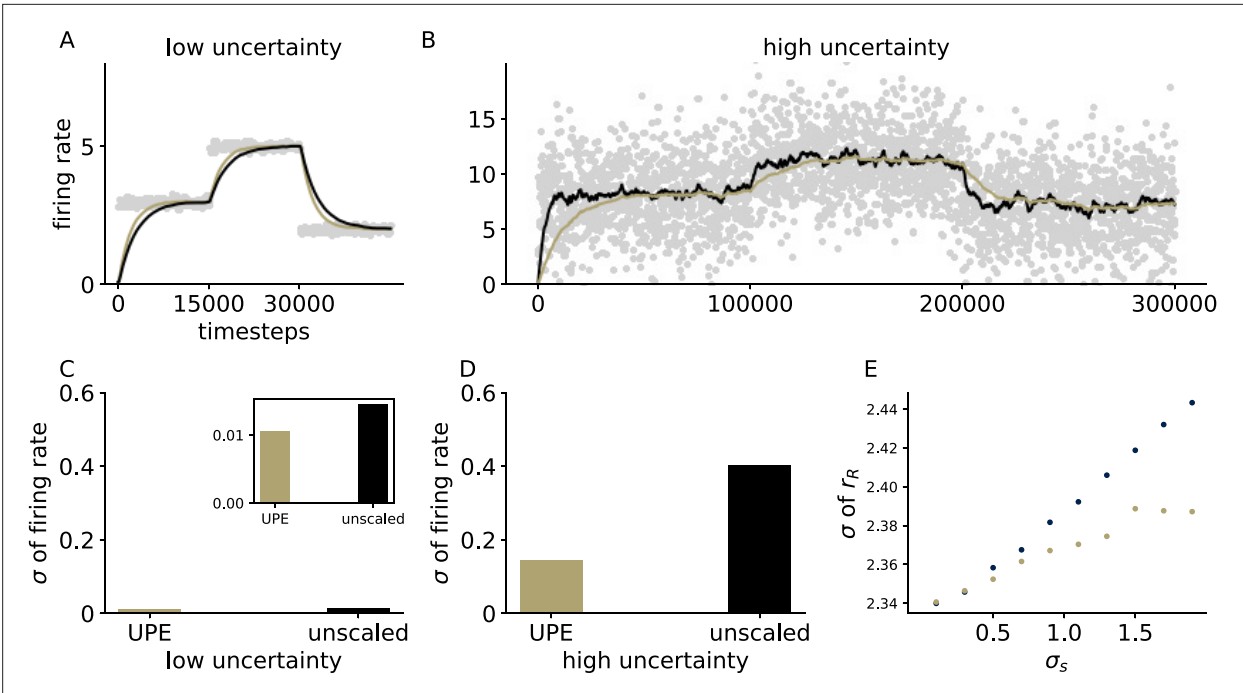

**Figure 7.** Effective learning rate is automatically adjusted with UPEs. (**A, B**) Firing rate over time of the representation neuron in a circuit with uncertainty-modulated prediction errors (gold) and in a circuit with unmodulated errors (black) in a low uncertainty setting (**A**) and a high uncertainty setting (**B**). (**C**) Standard deviation of the firing rate of the representation neuron in the low uncertainty setting (inset has a different scale, outer axis scale matches the one in **D**). (**D**) Standard deviation of the firing rate of the representation neuron in the high uncertainty setting. (**E**) Standard deviation of the firing rate $r_R$ as a function of the standard deviation of the presented stimulus distribution $\sigma_s$. Standard deviations were computed over 100,000 data points from timestep 100,000–200,000.

represent the predicted mean of the associated whisker deflections. Our model predicts that upon presentation of an unexpected whisker stimulus, those internal representation neurons adjust their activity to represent the new whisker deflection depending on the variability of the associated whisker deflections: they adjust their activity more (given equal deviations from the mean) if the associated whisker deflections are less variable (see the next section and *Figure 7*).

The following experimental results are compatible with our predictions: First, putative inhibitory neurons (narrow spiking units) in the macaque anterior cingulate cortex increased their firing rates in periods of high uncertainty (*Banaie Boroujeni et al., 2021*). These inhibitory neurons could correspond to the PVs in our model. Second, prediction error activity seems to decrease in less predictable, and hence more uncertain, contexts: in mice reared in a predictable environment [where locomotion and visual flow match (*Keller et al., 2012*)], error neuron responses to mismatches in locomotion and visual flow decreased with each day of experiencing these unpredictable mismatches. Third, the responses of SSTs and PVs to mismatches between locomotion and visual flow (*Attinger et al., 2017*) are in line with our model (note that in this experiment the mismatches are negative prediction errors as visual flow was halted despite ongoing locomotion): In this study, SST responses decreased during mismatch, that is when the visual flow was halted, and there was no difference between mice reared in a predictable or unpredictable environment. In line with these observations, the authors concluded that SST responses reflected the actual visual input. In our model negative PE circuit, SSTs also reflect the actual stimulus input, which in our case was a whisker stimulus (SST rates in *Figure 6C, I* reflect the stimuli (black and grey bar) in A and G, respectively) and SST rates are the same for high and low uncertainty (corresponding to mice reared in a predictable or unpredictable environment). In the same study, PV responses were absent towards mismatches in animals reared in an unpredictable environment (*Attinger et al., 2017*). The authors argued that mice reared in an unpredictable environment did not learn to form a prediction. In our model, the missing prediction corresponds to missing predictive input from the auditory domain (e.g. due to undeveloped synapses from the predictive auditory

input). If we removed the predictive input in our model, PVs in the negative PE circuit would also be silent as they would not receive any of the excitatory predictive inputs.

## The effective learning rate is automatically adjusted with UPEs

To test whether UPEs can automatically adjust the effective learning rate of a downstream neural population, we looked at two contexts that differed in uncertainty and compared how the mean representation evolves with and without UPEs. Indeed, in a low-uncertainty setting, a new mean representation can be learned faster with UPEs (in comparison to unmodulated, *Figure 7A and C*). In a high-uncertainty setting, the effective learning rate is smaller, and the mean representation is less variable than in the unmodulated case (*Figure 7B and D*). The standard deviation of the firing rate increases only sublinearly with the standard deviation of the inputs (*Figure 7E*). In summary, uncertainty-modulation of prediction errors enables an adaptive learning rate modulation.

## UPEs ensure uncertainty-based weighting of prior and sensory information

Behavioural studies suggest that during perception humans integrate priors or predictions ($p$) and sensory information ($s$) in a Bayes-optimal manner (*Ashourian and Loewenstein, 2011*; *Petzschner and Glasauer, 2011*). This entails that an internal neural representation ($r$, which determines perception, is achieved by weighting the two according to their uncertainties:

$$r = \frac{1}{c} \left( \frac{1}{\sigma_s^2} s + \frac{1}{\sigma_p^2} p \right),$$

(4)

where $c = \frac{1}{\sigma_s^2} + \frac{1}{\sigma_p^2}$, $\sigma_s^2$ is the uncertainty of the sensory information, and $\sigma_p^2$ is the uncertainty of the prior.

To obtain this weighting in the steady state, prediction errors from the lower area, the sensory prediction error ($s - r$), and from the local area, the representation prediction error ($r - p$), can be used to update the current representation [as in *Rao and Ballard, 1999*]. Maximising the log-likelihood (*Equation 22* in Methods and *Equation 35* in SI) yields an update of the representation by the difference between the bottom-up and top-down prediction errors.

$$\dot{r} = \frac{1}{\sigma_s^2}(s - r) - \frac{1}{\sigma_p^2}(r - p)$$

(5)

From this we obtain *Equation 4* by setting $\dot{r} = 0$. Translating the update in *Equation 5* into our framework of UPE circuits reads as

$$\dot{r} = (\text{UPE}_s^+ - \text{UPE}_s^-) - (\text{UPE}_p^+ - \text{UPE}_p^-),$$

(6)

illustrated in *Figure 8A*.

In our circuit, the representation neuron (green) receives the bottom-up errors, which are modulated by sensory uncertainty $\sigma_s^2$, as in the recurrent circuit model from the previous section. In addition, it receives errors between its current activity and a prior (a higher level expectation of the representation). The prior is set to the learned mean of the stimulus distribution and the errors are modulated by the learned prior uncertainty $\sigma_p^2$. We simulated this circuit, to which we presented stimuli drawn from Gaussian distributions as before. We varied the PV activity reflecting the sensory and prior uncertainty, and measured the activity of the representation neuron $r$ to each stimulus. The higher the uncertainty of the prior information $\sigma_s$, the more the representation reflects the current stimulus (*Figure 8B*, see also *Equation 4*). The higher the uncertainty of the sensory information $\sigma_s$, the more the representation reflects the prior mean (*Figure 8C*). This is a common behavioural effect, which is often referred to as central tendency effect or contraction bias (*Hollingworth, 1910*; *Jazayeri and Shadlen, 2010*; *Akrami et al., 2018*; *Meirhaeghe et al., 2021*).

To give a simple example how the prior uncertainty could come about in a dynamical environment, imagine noisy Gaussian sensory inputs with a fixed variance $\sigma_s^2$, and step changes in the mean, as in *Figure 7*. On the lowest level 0, after faithful learning, the learned variance $\sigma_0^2$ will represent the variance of the sensory input $\sigma_s^2$ for a given mean. If in addition, the mean of the sensory input varies (as in

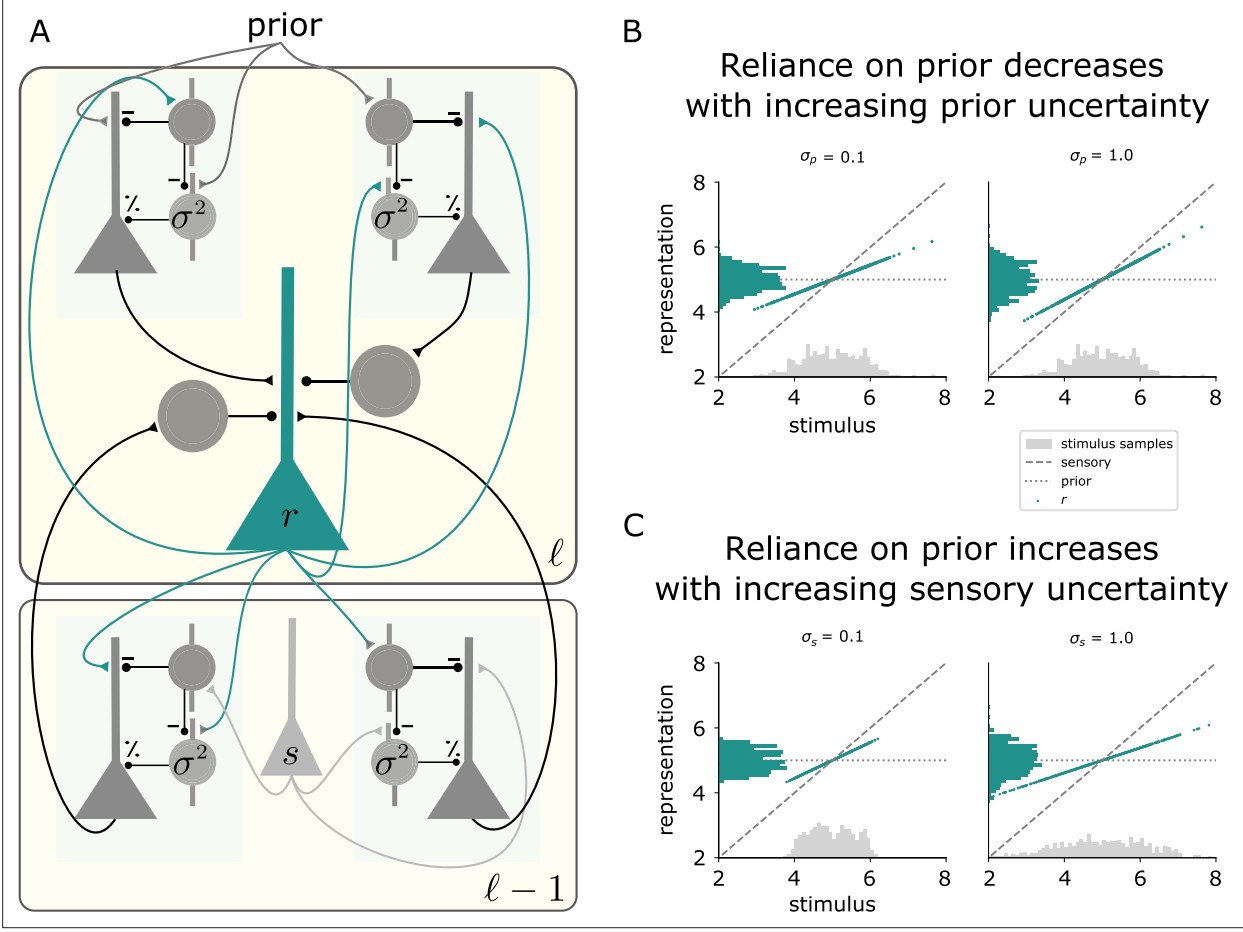

**Figure 8.** UPEs ensure uncertainty-based weighting of prior and sensory information. (**A**) Illustration of the hierarchical two-area model. A representation neuron ($r$) in area $\ell$ receives positive and negative UPEs from the area $\ell - 1$ below (sensory prediction error, as in **Figure 5**), and positive and negative UPEs from the same area (representation prediction error) with different signs, see **Equation 5**. In this example, the uncertainty in area $\ell - 1$ corresponds to the sensory uncertainty $\sigma_s^2$, and the uncertainty in the area $\ell$ above, corresponds to the prior uncertainty $\sigma_p^2$. Both uncertainties are represented by PV activity in the respective area. (**B, C**) The simulation results from the hierarchical circuit model yield an uncertainty-based weighting of prior and sensory information as in **Equation 4**. (**B**) The activity $r$ of the representation neuron as a function of the stimulus $s$ for different amounts of prior uncertainty $\sigma_p^2$, when the sensory uncertainty is fixed $\sigma_s^2 = 0.5$. The histograms show the distribution of the sensory stimulus samples (grey) and the distribution of the activity of the representation neuron (green). (**C**) The activity of the representation neuron as a function of the stimulus for different amounts of sensory uncertainty $\sigma_s^2$, when the prior uncertainty is fixed $\sigma_p^2 = 0.5$.

**Figure 7**), then on the level above $\sigma_1^2$ will reflect how much the mean varies. The former corresponds to the sensory uncertainty, the latter to the environmental volatility, which increases the uncertainty about the prediction for the mean ($\sigma_p^2$).

## Discussion

Based on normative theories, we propose that the brain uses uncertainty-modulated prediction errors. In particular, we hypothesise that layer 2/3 prediction error neurons represent prediction errors that are inversely modulated by uncertainty. Here, we showed that different inhibitory cell types in layer 2/3 cortical circuits can compute means and variances and thereby enable pyramidal cells to represent uncertainty-modulated prediction errors. We further showed that the cells in the circuit are able to learn to predict the means and variances of their inputs with local activity-dependent plasticity rules. Our study makes experimentally testable predictions for the activity of different cell types, PV and SST interneurons, in particular, prediction error neurons and representation neurons. Finally, we showed that circuits with uncertainty-modulated prediction errors enable adaptive learning rates, resulting in

fast learning when uncertainty is low and slow learning to avoid detrimental fluctuations when uncertainty is high.

Our theory has the following notable implications: The first implication concerns the hierarchical organisation of the brain. At each level of the hierarchy, we find similar canonical circuit motifs that receive both feedforward (from a lower level) and feedback (from a higher level, predictive) inputs that need to be integrated. We propose that uncertainty is computed on each level of the hierarchy. This enables uncertainty estimates specific to the processing level of a particular area. Experimental evidence is so far insufficient to favour distributed uncertainty representation over the idea that uncertainty is only computed on the level of decisions in higher level brain areas such as the parietal cortex (*Kiani and Shadlen, 2009*), orbitofrontal cortex (*Masset et al., 2020*), or prefrontal cortex (*Rushworth and Behrens, 2008*). Our study provides a concrete suggestion for a canonical circuit implementation and, therefore, experimentally testable predictions. The distributed account has clear computational advantages for task-flexibility, information integration, active sensing, and learning (see *Koblinger et al., 2021* for a recent review of the two accounts). Additionally, adding uncertainty-modulated prediction errors from different hierarchical levels according to the predictive coding model (*Rao and Ballard, 1999*; *Whittington and Bogacz, 2017*) yields an uncertainty-based weighting of feedback and feedforward information, similar to a Bayes-optimal weighting, which can be reconciled with human behaviour (*Payzan-LeNestour and Bossaerts, 2011*). Two further important implications result from storing uncertainty in the afferent connections to the PVs. First, this implies that the same PV cell can store different uncertainties depending on the context, which is encoded in the pre-synaptic activation. Second, fewer PVs than pyramidal cells are required for the mechanism, which is compatible with the 80/20 ratio of excitatory to inhibitory cells in the brain. The lower selectivity of interneurons in comparison to pyramidal cells could be a feature in prediction error circuits. Error neurons selective to similar stimuli are more likely to receive similar stimulus information, and hence similar predictions. Therefore, a circuit structure may have developed such that prediction error neurons with similar selectivity may receive inputs from the same inhibitory interneurons.

We claim that the uncertainty represented by PVs in our theoretical framework corresponds to *expected uncertainty* that results from noise or irreducible uncertainty in the stimuli and should therefore decrease the learning rate. Another common source of uncertainty are changes in the environment, also referred to as the *unexpected uncertainty*. In volatile environments with high unexpected uncertainty, the learning rate should increase. We suggest that vasointestinal-peptide-positive interneurons (VIPs) could be responsible for signalling the unexpected uncertainty, as they respond to reward, punishment and surprise (*Pi et al., 2013*), which can be indicators of high unexpected uncertainty. They provide potent disinhibition of pyramidal cells (*Pfeffer, 2014*), and also inhibit PVs in layer 2/3 (*Pfeffer et al., 2013*). Hence, they could increase error activity resulting in a larger learning signal. In general, interneurons are innervated by different kinds of neuromodulators (*Lee et al., 2013*; *Prönneke et al., 2015*) and control pyramidal cell's activity and plasticity (*Isaacson and Scanziani, 2011*; *Gidon and Segev, 2012*; *Wilmes et al., 2016*; *Wilmes et al., 2017*; *Wilmes and Clopath, 2019*). Therefore, neuromodulators could have powerful control over error neuron activity and hence perception and learning.

A diversity of proposals about the neural representation of uncertainty exist. For example, it has been suggested that uncertainty is represented in single neurons by the width (*Fischer and Peña, 2011*), or amplitude of their responses (*Ma et al., 2006*), or implicitly via sampling [neural sampling hypothesis; (*Petrovici et al., 2013*; *Buesing et al., 2011*; *Berkes et al., 2011*)], or rather than being represented by a single feature, can be decoded from the activity of an entire population (*Dehaene et al., 2021*). While we suggest that PVs represent uncertainty to modulate prediction error responses, we do not claim that this is the sole representation of uncertainty in neuronal circuits.

Uncertainty estimation is relevant for Bayes-optimal integration of different sources of information, for example different modalities [multi-sensory integration; *Ernst and Banks, 2002*; *Fetsch et al., 2012*] or priors and sensory information. Here, we present a circuit implementation for weighting sensory information versus priors according to their uncertainties by integrating uncertainty-modulated prediction errors. An alternative solution is to estimate uncertainty from the activity of prediction error neurons and use it to weight priors and sensory information (*Hertäg et al., 2023*), leading to contraction bias (*Hertäg and Clopath, 2022*). *Hertäg and Clopath, 2022* previously showed that the integration of prediction errors with sensory information in representation neurons can also lead

to contraction bias, but without being dependent on uncertainty. Instead of modulating the error on each level by the uncertainty on that level as in our suggestion, one can also obtain a Bayes-optimal weighting by combining an unweighted top-down error with a bottom-error that is multiplicatively modulated by the prior uncertainty, and divisively modulated by the bottom-up uncertainty (*Granier et al., 2024*). It has also been suggested that Bayes-optimal multi-sensory integration could be achieved in single neurons (*Fetsch et al., 2012*; *Jordan et al., 2022*). Our proposal is complementary to these solutions in that uncertainty-modulated errors can be forwarded to other cortical and subcortical circuits at different levels of the hierarchy, where they can be used for inference and learning. It further allows for a context-dependent integration of sensory inputs.

Multiple neurological disorders, such as autism spectrum disorder or schizophrenia, are associated with maladaptive contextual uncertainty-weighting of sensory and prior information (*Goris et al., 2021*; *Lawson et al., 2014*; *Van de Cruys et al., 2014*; *Lawson et al., 2017*; *Shi et al., 2022*). These disorders are also associated with aberrant inhibition, for example ASD is associated with an excitation-inhibition imbalance (*Rubenstein and Merzenich, 2003*) and reduced inhibition (*Harada et al., 2011*; *Gaetz et al., 2014*). Interestingly, PV cells, in particular chandelier PV cells, were shown to be reduced in number and synaptic strength in ASD (*Juarez and Martínez Cerdeño, 2022*). Our theory provides one possible explanation of how deficits in uncertainty-weighting on the behavioural level could be linked to altered PVs on the circuit level.

Finally, uncertainty-modulated errors could advance deep hierarchical neural networks. In addition to propagating gradients, propagating uncertainty may have advantages for learning. The additional information on uncertainty could enable calculating distances between distributions, which can provide an informative and parameter-independent metric for learning [e.g. natural gradient learning (*Kreutzer et al., 2022*)].

To provide experimental predictions that are immediately testable, we suggested specific roles for SSTs and PVs, as they can subtractively and divisively modulate pyramidal cell activity, respectively. In principle, our theory more generally posits that any subtractive or divisive inhibition could implement the suggested computations. With the emerging data on inhibitory cell types, subtypes of SSTs and PVs or other cell types may turn out to play the proposed role.

The model predicts that the divisive interneuron type, which we here suggest to be the PVs, receives a representation of the stimulus as an input. PVs could be pooling the inputs from stimulus-responsive layer 2/3 neurons to estimate uncertainty. The more the stimulus varies, the larger the variability of the pyramidal neuron responses and, hence, the variability of the PV activity. The broader sensory tuning of PVs (*Cottam et al., 2013*) is in line with the model insofar as uncertainty modulation could be more general than the specific feature, which is more likely for low-level features processed in primary sensory cortices. PVs were shown to connect more to pyramidal cells with similar feature tuning (*Znamenskiy et al., 2024*) this would be in line with the model, as uncertainty modulation should be feature-related. In our model, some SSTs deliver the prediction to the positive prediction error neurons. SSTs are already known to be involved in spatial prediction, as they underlie the effect of surround suppression (*Adesnik et al., 2012*), in which SSTs suppress the local activity dependent on a predictive surround.

In the model we propose, SSTs should be subtractive and PVs divisive. However, SSTs can also be divisive, and PVs subtractive dependent on circuit and postsynaptic properties (*Seybold et al., 2015*; *Lee et al., 2012*; *Dorsett et al., 2021*). This does not necessarily contradict our model, as circuits in which SSTs are divisive and PVs subtractive could implement a different function, as not all pyramidal cells are error neurons. Hence, our model suggests that error neurons which can calculate UPEs should have similar physiological properties to the layer 2/3 cells observed in the study by *Wilson et al., 2012*.

Our model further posits the existence of two distinct subtypes of SSTs in positive and negative error circuits. Indeed, there are many different subtypes of SSTs (*Schneider-Mizell et al., 2024*). SST is expressed by a large population of interneurons, which can be further subdivided. There is, for example, a type called SST44, which was shown to specifically respond when the animal corrects a movement (*Green et al., 2023*). Our proposal is hence aligned with the observation of functionally specialised subtypes of SSTs. Importantly, the comparison between stimulus and prediction needs to happen before the divisive modulation. Although our model does not make assumptions about the precise dendritic location of this comparison, we suggest this to happen on the apical dendrite, as

top-down inputs and SST synapses arrive there. SSTs receive top-down inputs (*Liu et al., 2020*), which could provide the prediction to be subtracted in negative prediction error circuits.

To compare predictions and stimuli in a subtractive manner, the encoded prediction/stimulus needs to be translated into a direct variable code. In this framework, we assume that this can be achieved by the weight matrix defining the synaptic connections from the neural populations representing predictions and stimuli (possibly in a population code).

To enable the comparison between predictions and sensory information via subtractive inhibition, we pointed out that the weights of those inputs on the postsynaptic neuron need to match. This essentially means that there needs to be a balance of excitatory and inhibitory inputs. Such an EI balance has been observed experimentally (*Tan and Wehr, 2009*). And it has previously been suggested that error responses are the result of breaking this EI balance (*Hertäg and Sprekeler, 2020*; *Barry and Gerstner, 2024*). Heterosynaptic plasticity is a possible mechanism to achieve EI balance (*Field et al., 2020*). For example, spike pairing in pre- and postsynaptic neurons induces long-term potentiation at co-activated excitatory and inhibitory synapses with the degree of inhibitory potentiation depending on the evoked excitation (*D'amour and Froemke, 2015*), which can normalise EI balance (*Field et al., 2020*).

## Conclusion

To conclude, we proposed that prediction error activity in layer 2/3 circuits is modulated by uncertainty and that the diversity of cell types in these circuits achieves the appropriate scaling of the prediction error activity. The proposed model is compatible with Bayes-optimal behaviour and makes predictions for future experiments.

## Methods

### Derivation of the UPE

The goal is to learn $\hat{\mu}$ to maximise the log-likelihood:

$$\log \mathcal{L} = \log p(s|\hat{\mu}, \sigma)$$

$$= \log \prod_{n=1}^{N} \mathcal{N}(s_n|\hat{\mu}, \sigma)$$

$$= -\frac{1}{2\sigma^2} \sum_{n=1}^{N}(s_n - \hat{\mu})^2 - \frac{N}{2}\log(2\pi\sigma^2)$$

We consider the log-likelihood for one sample $s$ of the stimulus distribution:

$$\log p(s|\hat{\mu}, \sigma) = -\frac{1}{2\sigma^2}(s - \hat{\mu})^2 - \frac{1}{2}\log(2\pi\sigma^2)$$

Stochastic gradient ascent on the log-likelihood gives the update for $\hat{\mu}$:

$$\Delta\hat{\mu} \propto \frac{\partial}{\partial\hat{\mu}}(\log p(s|\hat{\mu}, \sigma))$$

$$= \frac{\partial}{\partial\hat{\mu}}\left(-\frac{1}{2\sigma^2}(s - \hat{\mu})^2 - \frac{1}{2}\log(2\pi\sigma^2)\right)$$

$$= \frac{1}{\sigma^2}(s - \hat{\mu}).$$

### Circuit model

#### Prediction error circuit

We modelled a circuit consisting of excitatory prediction error neurons in layer 2/3, and two inhibitory populations, corresponding to PV and SST interneurons.

Layer 2/3 pyramidal cells receive divisive inhibition from PVs (*Wilson et al., 2012*). We, hence, modelled the activity of prediction error neurons as

**Table 1.** Parameters of the network.

| Parameter | Value | Description |
|---|---|---|
| $w_{\mathrm{PV}^+,\mathrm{SST}^+}$ | $\sqrt{\frac{2-\beta}{\beta}}$ | Weight from $\mathrm{SST}^+$ to $\mathrm{PV}^+$ |
| $w_{\mathrm{PV}^+,\mathrm{s}}$ | $\sqrt{\frac{2-\beta}{\beta}}$ | Weight from s to $\mathrm{PV}^+$ |
| $w_{\mathrm{PV}^-,\mathrm{SST}^-}$ | $\sqrt{\frac{2-\beta}{\beta}}$ | Weight from $\mathrm{SST}^-$ to $\mathrm{PV}^-$ |
| $w_{\mathrm{PV}^-,\mathrm{R}}$ | $\sqrt{\frac{2-\beta}{\beta}}$ | Weight from R to $\mathrm{PV}^-$ |
| $w_{\mathrm{SST}^+,\mathrm{R}}$ | 1.0 | Weight from R to $\mathrm{SST}^+$ |
| $w_{\mathrm{SST}^-,\mathrm{s}}$ | 1.0 | Weight from s to $\mathrm{SST}^-$ |
| $w_{\mathrm{UPE}^+,\mathrm{SST}^+}$ | 1.0 | Weight from $\mathrm{SST}^+$ to $\mathrm{UPE}^+$ |
| $w_{\mathrm{UPE}^+,\mathrm{s}}$ | 1.0 | Weight from s to $\mathrm{UPE}^+$ |
| $w_{\mathrm{UPE}^-,\mathrm{R}}$ | 1.0 | Weight from R to $\mathrm{UPE}^-$ |
| $w_{\mathrm{UPE}^-,\mathrm{SST}^-}$ | 1.0 | Weight from $\mathrm{SST}^-$ to $\mathrm{UPE}^-$ |
| $w_{\mathrm{R,UPE}^+}$ | 0.1(1.0) | Weight from $\mathrm{UPE}^+$ to R (*Figure 6*) |
| $w_{\mathrm{R,UPE}^-}$ | 0.1(1.0) | Weight from $\mathrm{UPE}^-$ to R (*Figure 6*) |
| $x_{\max}$ | 20 | Limits neuronal activity |
| $\beta$ | 0.1 | Nudging parameter |

$$\tau_{\mathrm{E}}\frac{dr_{\mathrm{UPE}}}{dt} = -r_{\mathrm{UPE}} + \phi\left(\frac{I_{\mathrm{dend}}}{I_0 + w_{\mathrm{UPE,PV}}\, r_{\mathrm{PV}}}\right), \tag{7}$$

where $\phi(x)$ is the activation function defined as:

$$\phi(x) = \begin{cases} 0 & \text{if } x \leq 0 \\ x & \text{if } 0 < x < x_{\max} \\ r_{\max} & \text{if } x \geq x_{\max}, \end{cases} \tag{8}$$

$I_{\mathrm{dend}} = \lfloor w_{\mathrm{UPE,s}}\, \mathrm{s} - w_{\mathrm{UPE,SST}}\, r_{\mathrm{SST}} \rfloor^k$ is the dendritic input current to the positive prediction error neuron (see section Neuronal dynamics below for $r_x$ and for the negative prediction error neuron, and *Table 1* for $w_x$). The nonlinearity in the dendrite is determined by the exponent $k$, which is by default $k = 2$, unless otherwise specified as in *Figure 4G–J*. $I_0 > 1$ is a constant ensuring that the divisive inhibition does not become excitatory, when $\sigma < 1.0$. All firing rates are rectified to ensure that they remain positive.

In the positive prediction error circuit, in which the SSTs learn to represent the mean, the SST activity is determined by

$$\tau_{\mathrm{I}}\frac{dr_{\mathrm{SST}^+}}{dt} = -r_{\mathrm{SST}^+} + \phi((1 - \beta)w_{\mathrm{SST}^+,\mathrm{a}}\, r_{\mathrm{a}} + \beta s). \tag{9}$$

**Table 2.** Additional parameters of the hierarchical network.

| Parameter | Value | Description |
|---|---|---|
| $w_{R,UPE_\ell^+}$ | 1.0 | Weight from $UPE_\ell^+$ to $R$ |
| $w_{R,UPE_\ell^-}$ | 1.0 | Weight from $UPE_\ell^-$ to $R$ |
| $w_{UPE_\ell^-,R}$ | 1.0 | Weight from $R$ to $UPE_\ell^-$ |
| $w_{UPE^-,prior}$ | 1.0 | Weight from $R$ to $UPE^-$ |

The SST activity is influenced (nudged with a factor $\beta$) by the somatosensory stimuli $s$, which provide targets for the desired SST activity. The connection weight from the sound representation to the SSTs $w_{SST,a}$ is plastic according to the following local activity-dependent plasticity rule (*Urbanczik and Senn, 2014*):

$$\Delta w_{SST,a} = \eta(r_{SST} - \phi(w_{SST,a}\ r_a))\ r_a, \tag{10}$$

where $\eta$ is the learning rate, $r_a$ is the pre-synaptic firing rate, $r_{SST}$ is the post-synaptic firing rate of the SSTs, $\phi(x)$ is a rectified linear activation function of the SSTs.

The learning rule ensures that the auditory input alone causes SSTs to fire at their target activity. As in the original proposal (*Urbanczik and Senn, 2014*), the terms in the learning rule can be mapped to local neuronal variables, which could be represented by dendritic ($w_{SST,a}\ r_a$) and somatic ($r_{SST}$) activity.

The PV firing rate is determined by the input from the sound representation ($w_{PV^+,a}\ r_a$) and the whisker stimuli, from which their mean is subtracted ($w_{PV^+,s}\ s - w_{PV^+,SST^+}\ r_{SST^+}$, where the mean is given by $r_{SST^+}$). The mean-subtracted whisker stimuli serve as a target for learning the weight from the sound representation to the PV $\eta$. The PV firing rate evoles over time according to:

$$\tau_I \frac{dr_{PV^+}}{dt} = -r_{PV^+} + \phi_{PV}((1-\beta)w_{PV^+,a}\ r_a + \beta(w_{PV^+,s}\ s - w_{PV^+,SST^+}\ r_{SST^+}))) \tag{11}$$

where $\phi_{PV}(x)$ is a rectified quadratic activation function, defined as follows:

$$\phi(x) = \begin{cases} 0 & \text{if } x \leq 0 \\ x^2 & \text{if } 0 < x < x_{max} \\ r_{max} & \text{if } x \geq x_{max}. \end{cases} \tag{12}$$

The connection weight from the sound representation to the PVs $w_{PV,a}$ is plastic according to the same local activity-dependent plasticity rule as the SSTs (*Urbanczik and Senn, 2014*):

$$w_{PV,a} = \eta(r_{PV} - \phi_{PV}(w_{PV,a}\ r_a))\ r_a. \tag{13}$$

The weight from the sound representation to the PV $w_{PV^+,a}$ approaches $\sigma$ (instead of μ as the weight to the SSTs), because the PV activity is a function of the mean-subtracted whisker stimuli (instead of the whisker stimuli as the SST activity), and for a Gaussian-distributed stimulus $s \sim \mathcal{N}(s|\mu,\sigma)$, it holds that $\mathbb{E}\left[\lfloor s - \mu \rfloor^+\right] \propto \sigma$.

**Table 3.** Inputs.

| Parameter | Value | Description |
|---|---|---|
| a | {0.0, 1.0} | Auditory stimulus (on/off) |
| s | $\sim \mathcal{N}(\mu,\sigma)$ | Somatosensory (whisker) stimulus |
| N | 1000–20,000 | Number of whisker stimulus samples |
| D | {10, 100} | Stimulus duration (*Figures 1–5 and 7*) |

**Table 4.** Parameters of the plasticity rules.

| Parameter | Value | Description |
|---|---|---|
| $\eta_{SST}$ | 0.1 | Learning rate for $w_{SST^{+/-},a}$ |
| $\eta_{PV}$ | $0.01 * \eta_R = 0.001$ | Learning rate for $w_{PV^{+/-},a}$ |
| $\eta_R$ | 0.1 | Learning rate for $w_{R,a}$ |
| $w_{SST,a}^{initial}$ | 0.01 | Initial value for $w_{SST^{+/-},a}$ |
| $w_{PV,a}^{initial}$ | 0.01 | Initial value for $dt = 0.1$ |
| $w_{R,a}^{initial}$ | 0.01 | Initial value for $w_{R,a}$ |

## Recurrent circuit model

In the recurrent circuit, shown in *Figure 5*, we added an internal representation neuron to the circuit with firing rate $r_R$. In this circuit, the SSTs in the positive PE circuit inherit the mean representation from the representation neuron instead of learning it themselves, that is why they receive an input $w_{PV^+,a}$. The SSTs in the negative circuit inherit the stimulus representation and hence receive an input $w_{SST^-,s}$ s In this recurrent circuit, the firing rate of each population $r_i$ where $i \in [SST^+, SST^-, PV^+, PV^-, UPE^+, UPE^-, R]$ evolves over time according to the following neuronal dynamics. $\phi$ denotes a rectified linear activation function with saturation, $\phi_{PV}$ denotes a rectified quadratic activation function with saturation, defined in the section below. All firing rates are rectified to ensure that they remain positive.

$$\tau_I \frac{dr_{SST^+}}{dt} = -r_{SST^+} + \phi(w_{SST^+,R}\ r_R), \tag{14}$$

$$\tau_I \frac{dr_{PV^+}}{dt} = -r_{PV^+} + \phi_{PV}((1-\beta)w_{PV^+,a}\ r_a + \beta(w_{PV^+,s}\ s - w_{PV^+,SST^+}\ r_{SST^+}))), \tag{15}$$

$$\tau_E \frac{dr_{UPE^+}}{dt} = -r_{UPE^+} + \phi\left(\frac{\lfloor w_{UPE,s}\ s - w_{UPE,SST^+}\ r_{SST^+}\rfloor^k}{I_0 + w_{UPE,PV^+}\ r_{PV^+}}\right), \tag{16}$$

$$\tau_I \frac{dr_{SST^-}}{dt} = -r_{SST^-} + \phi(w_{SST^-,s}\ s), \tag{17}$$

$$\tau_I \frac{dr_{PV^-}}{dt} = -r_{PV^-} + \phi_{PV}((1-\beta)w_{PV^-,a}\ r_a + \beta(w_{PV^-,R}\ r_R - w_{PV^-,SST^-}\ r_{SST^-}))), \tag{18}$$

$$\tau_E \frac{dr_{UPE^-}}{dt} = -r_{UPE^-} + \phi\left(\frac{\lfloor w_{UPE,R}\ r_R - w_{UPE,SST^-}\ r_{SST^-}\rfloor^k}{I_0 + w_{UPE,PV^-}\ r_{PV^-}}\right), \tag{19}$$

$$\tau_E \frac{dr_R}{dt} = -r_R + \phi(w_{R,a}\ r_a + w_{R,UPE^+}\ r_{UPE^+} - w_{R,UPE^-}\ r_{UPE^-})). \tag{20}$$

## Hierarchical predictive coding

The idea behind hierarchical predictive coding is that the brain infers or represents the causes of its sensory inputs using a hierarchical generative model (*Friston, 2005*). Each level of the cortical hierarchy provides a prior for the mean of the lower level representation, with the top level representation $r_L$ being determined by the context.

**Table 5.** Parameters of simulations in *Figures 2–5*.

| Parameter | Value | Description |
|---|---|---|
| $T$ | $N * D$ | Simulation time |
| $dt$ | 0.1 | Simulation time step |
| $\tau_E$ | 1.0 | Excitatory membrane time constant |
| $\tau_I$ | 1.0 | Inhibitory membrane time constant |

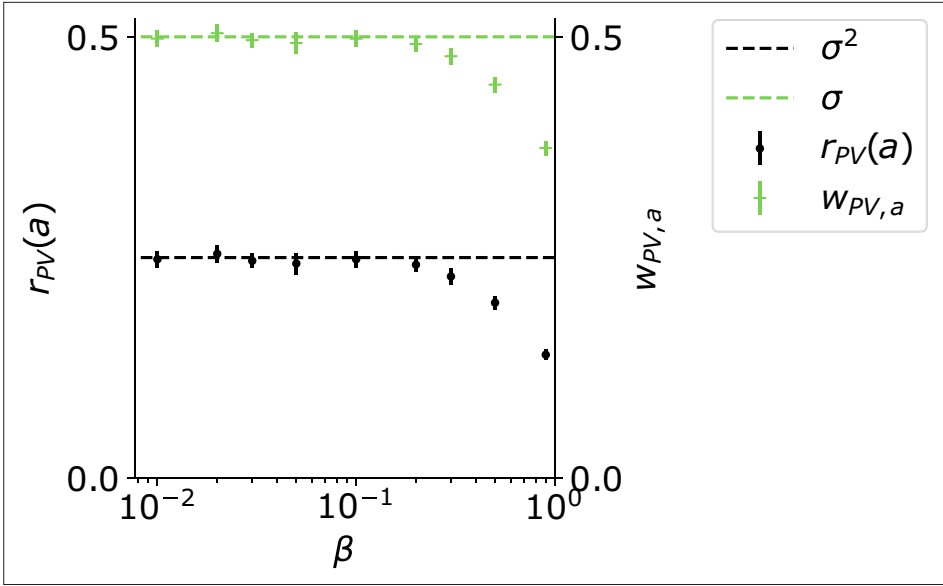

**Figure 9.** For small $\beta$, and $w_s = \sqrt{\frac{2-\beta}{\beta}}$, the weight from $a$ to PV approaches $\sigma$ and the PV firing rate approaches $\sigma^2$.

Noise enters in the sensory area by sampling a stimulus $s = r_0$. In the sensory area, the variance $\sigma_0^2$ is learned to match the variability of the external stimulus $\sigma_s^2$.

The goal is to infer the set of latent representations $\mathbf{r} = (r_1, .., r_{L-1})$, given the synaptic weights $w_\ell$ of the internal model and the top activity $r_L$, that best reproduces the sensory inputs $s$. To this end, we minimise the following energy:

$$E = \frac{1}{2} \sum_{\ell=0}^{L-1} \frac{1}{\sigma_\ell^2} \| r_\ell - \rho(w_\ell r_{\ell+1}) \|_2^2 \tag{21}$$

where $\rho$ is a transfer function.

We obtain the update for $r_\ell$ with gradient descent on the energy with respect to $r_\ell$:

$$
\begin{aligned}
\dot{r}_\ell \quad &= -\frac{\partial E}{\partial r_\ell} = \frac{1}{\sigma_{\ell-1}^2}(r_{\ell-1} - \rho(w_{\ell-1} r_\ell))w_{\ell-1}\rho'(r_\ell) - \frac{1}{\sigma_\ell^2}(r_\ell - \rho(w_\ell r_{\ell+1})) \\
&= \frac{1}{\sigma_{\ell-1}^2}\rho'(w_{\ell-1} r_\ell)e_{\ell-1} - \frac{1}{\sigma_\ell^2}e_\ell
\end{aligned}
\tag{22}
$$

In our model, $w_\ell$ are scalars as they denote single weights.

We obtain the steady-state representation $r_\ell$ by setting its derivative in *Equation 21* to 0:

$$0 \overset{!}{=} \frac{w_{\ell-1}\rho'(w_{\ell-1} r_\ell)}{\sigma_{\ell-1}^2}(r_{\ell-1} - \rho(w_{\ell-1} r_\ell)) - \frac{1}{\sigma_\ell^2}(r_\ell - \rho(w_\ell r_{\ell+1})) \tag{23}$$

**Table 6.** Parameters of the simulation in *Figure 6*.

| Parameter | Value | Description |
|---|---|---|
| $T$ | $N * D$ | Simulation time |
| $dt$ | 0.1 | Simulation time step |
| $\tau_E$ | 10.0 | Excitatory membrane time constant |
| $\tau_I$ | 2.0 | Inhibitory membrane time constant |
| $\eta_R$ | 0.01 | Learning rate of $w_{R,A}$ |

We next consider, for simplicity, a threshold-linear transfer function $\rho(w_{\ell-1}r_\ell)$ such that if $w_{\ell-1}r_\ell < 0$, then $\rho(w_{\ell-1}r_\ell) = 0$ and its derivative $\rho' = 0$ or if $w_{\ell-1}r_\ell \geq 0$ then $\rho(w_{\ell-1}r_\ell) = w_{\ell-1}r_\ell$ and $\rho' = 1$.

Solving **Equation 24** for $r_\ell$ and assuming $r_\ell \geq 0$, we get:

$$r_\ell = \frac{\frac{w_{\ell-1}^2}{\sigma_{\ell-1}^2}}{\frac{w_{\ell-1}^2}{\sigma_{\ell-1}^2} + \frac{1}{\sigma_\ell^2}} \frac{r_{\ell-1}}{w_{\ell-1}} + \frac{\frac{1}{\sigma_\ell^2}}{\frac{w_{\ell-1}^2}{\sigma_{\ell-1}^2} + \frac{1}{\sigma_\ell^2}} w_\ell r_{\ell+1}. \tag{24}$$

See appendix for the general case with any transfer function and weight matrix.

## Hierarchical circuit model

In the hierarchical circuit model, the representation neuron does not only receive UPEs from the area below, but also from the current area.

$$\tau_E \frac{dr_R}{dt} = -r_R + \phi(w_{R,a}\, r_a + w_{R,UPE_{\ell-1}^+}\, r_{UPE_{\ell-1}^+} - w_{R,UPE_{\ell-1}^-}\, r_{UPE_{\ell-1}^-} - w_{R,UPE_\ell^+}\, r_{UPE_\ell^+} + w_{R,UPE_\ell^-}\, r_{UPE_\ell^-}). \tag{25}$$

The UPEs from the area below are as defined in the recurrent circuit model, and the UPEs from the current area are defined accordingly as:

$$\tau \frac{dr_{UPE_\ell^+}}{dt} = -r_{UPE_\ell^+} + \phi\left(\frac{\lfloor w_{UPE,r_R} - w_{UPE,SST^+}\, r_{SST^+}\rfloor^k}{I_0 + w_{UPE,PV^+}\, r_{PV^+}}\right), \tag{26}$$

$$\tau_E \frac{dr_{UPE_\ell^-}}{dt} = -r_{UPE_\ell^-} + \phi\left(\frac{\lfloor w_{UPE,prior}\, \mu_{prior} - w_{UPE,SST^-}\, r_{SST^-}\rfloor^k}{I_0 + w_{UPE,PV^-}\, r_{PV^-}}\right), \tag{27}$$

The computations and parameters in each area are the same as for the recurrent circuit model above and in **Figure 5**.

Synapses from the higher level representation of the sound $a$ to R were plastic according to the following activity-dependent plasticity rules (**Urbanczik and Senn, 2014**).

$$\Delta w_{R,a} = \eta(r_R - \phi(w_{R,a}r_a))r_a, \tag{28}$$

where $\eta_{PV} = 0.01\eta_R$.

## Estimating the variance correctly

The PVs estimate the variance of the sensory input from the variance of the teaching input $(s - \mu)$, which nudges the membrane potential of the PVs with a nudging factor $\beta$. The nudging factor reduces the effective variance of the teaching input, such that in order to correctly estimate the variance, this reduction needs to be compensated by larger weights from the SSTs to the PVs ($w_{PV,SST}$) and from the sensory input to the PVs ($w_{PV,s}$). To determine how strong the weights $w_s = w_{PV,SST} = w_{PV,s}$ need to be to compensate for the downscaling of the input variance by $\beta$, we require that $\mathbb{E}[wa]^2 = \sigma^2$ when the average weight change $\mathbb{E}[\Delta w] = 0$. The learning rule for $w$ is as follows:

$$\Delta w = \eta[r_{PV} - \phi(wa)]a$$
$$= \eta[\phi((1 - \beta)wa + \beta w_s \tilde{s}) - \phi(wa)]a$$

where $\tilde{s} = (s - \mu) \sim \mathcal{N}(0, \sigma)$ and $\tilde{s} = (s - \mu) \sim \mathcal{N}(0, \sigma)$.

Using that $\phi(u) = u^2$, the average weight change becomes:

$$\mathbb{E}[\Delta w] = \mathbb{E}[\eta((1 - \beta)^2 w^2 a^2 + \beta^2 w_s^2 \tilde{s}^2 + 2(1 - \beta)wa\beta w_s \tilde{s} - w^2 a^2)a]$$
$$= \mathbb{E}[\eta((1 + \beta^2 - 2\beta)w^2 a^2 + \beta^2 w_s^2 \tilde{s}^2 + 2(1 - \beta)wa\beta w_s \tilde{s} - w^2 a^2)a] \quad | \quad \mathbb{E}[\tilde{s}] = 0$$
$$= \mathbb{E}[\eta(\beta^2 w^2 a^2 - 2\beta w^2 a^2 + \beta^2 w_s^2 \tilde{s}^2)a]$$
$$= \mathbb{E}[\eta\beta(\beta w^2 a^2 - 2w^2 a^2 + \beta w_s^2 \tilde{s}^2)a]$$

$$= \eta\beta((\beta - 2)\mathbb{E}[(wa)^2] + \beta w_s^2 \mathbb{E}[\tilde{s}^2])a \quad | \quad \mathbb{E}[\tilde{s}^2] = \mathbb{E}[(s - \mu)^2] = \sigma^2$$

$$= \eta\beta((\beta - 2)\mathbb{E}[(wa)^2] + \beta w_s^2 \sigma^2)a$$

Given our objective $\mathbb{E}[(wa)^2] = \sigma^2$, we can write:

$$\mathbb{E}[\Delta w] = \eta\beta((\beta - 2)\sigma^2 + \beta w_s^2 \sigma^2)a$$

Then for $\mathbb{E}[\Delta w] = 0$:

$$0 = \beta - 2 + \beta w_s^2$$

$$\Rightarrow w_s = \sqrt{\frac{2 - \beta}{\beta}}$$

Here, we assumed that $\phi(u) = u^2$ instead of $\phi(u) = \lfloor u \rfloor^2$. To test how well this approximation holds, we simulated the circuit for different values of $\beta$ and hence $w_s$, and plotted the PV firing rate $r_{\mathrm{PV}}(a)$ given the sound input $a$ and the weight from $a$ to PV, $w_{\mathrm{PV},a}$, for different values of $\beta$ (*Figure 9*). This analysis shows that the approximation holds for small $\beta$ up to a value of $\beta = 0.2$.

We initialised the circuits with the initial weight configuration in *Table 1* and *Table 2*, and neural firing rates were initialised to be 0 ($r_i(0) = 0$ with $i \in [\mathrm{SST}^+, \mathrm{SST}^-, \mathrm{PV}^+, \mathrm{PV}^-, \mathrm{UPE}^+, \mathrm{UPE}^-, \mathrm{R}]$). We then paired a constant tone input with N samples from the whisker stimulus distribution, the parameters of which we varied and are indicated in each Figure. Each whisker stimulus intensity was presented for $D$ timesteps (see *Table 3*). Parameters of the plasticity rules can be found in *Table 4*. All simulations were written in Python. Differential equations were numerically integrated with a time step of $dt = 0.1$. Parameters of the simulations can be found in *Tables 5 and 6*.

Eliciting responses to mismatches (*Figure 4* and *Figure 6*).

We first trained the circuit with 10,000 stimulus samples to learn the variances in the a-to-PV weights. Then we presented different mismatch stimuli to calculate the error magnitude for each mismatch of magnitude $s - \mu$.

Comparing the UPE circuit with an unmodulated circuit (*Figure 7*).

To ensure a fair comparison, the unmodulated control has an effective learning rate that is the mean of the two effective learning rates in the uncertainty-modulated case.

## Acknowledgements

We thank Loreen Hertäg and Jakob Jordan for helpful discussions and Loreen Hertäg and Sadra Sadeh for feedback on the manuscript. This work has received funding from the European Union 7th Framework Programme under grant agreement 604102 (HBP), the Horizon 2020 Framework Programme under grant agreements 720270, 785907 and 945539 (HBP) and the Manfred Stärk Foundation.

## Additional information

### Funding

| Funder | Grant reference number | Author |
| --- | --- | --- |
| Horizon 2020 Framework Programme | 10.3030/720270 | Walter Senn<br>Mihai A Petrovici |
| Horizon 2020 Framework Programme | 10.3030/785907 | Walter Senn<br>Mihai A Petrovici |
| Horizon 2020 Framework Programme | 10.3030/945539 | Walter Senn<br>Mihai A Petrovici |
| European Union 7th Framework Programme | 604102 | Walter Senn |
| Horizon Europe Programme | 10.3030/101147319 | Walter Senn<br>Mihai A Petrovici |

| Funder | Grant reference number | Author |
|---|---|---|
| Manfred Stärk Stiftung | | Mihai A Petrovici |

The funders had no role in study design, data collection and interpretation, or the decision to submit the work for publication.

## Author contributions

Katharina Anna Wilmes, Conceptualization, Resources, Data curation, Software, Formal analysis, Validation, Investigation, Visualization, Methodology, Writing - original draft, Project administration, Writing - review and editing; Mihai A Petrovici, Formal analysis, Methodology, Writing - review and editing; Shankar Sachidhanandam, Resources, Data curation, Funding acquisition; Walter Senn, Formal analysis, Supervision, Funding acquisition, Writing - review and editing

## Author ORCIDs

Katharina Anna Wilmes ⓘ https://orcid.org/0000-0003-4948-1864
Mihai A Petrovici ⓘ https://orcid.org/0000-0003-2632-0427
Shankar Sachidhanandam ⓘ https://orcid.org/0000-0002-9359-6653
Walter Senn ⓘ https://orcid.org/0000-0003-3622-0497

Reviewer #2 (Public review): https://doi.org/10.7554/eLife.95127.4.sa1
Reviewer #4 (Public review): https://doi.org/10.7554/eLife.95127.4.sa2
Author response https://doi.org/10.7554/eLife.95127.4.sa3

# Additional files

## Supplementary files

MDAR checklist

## Data availability

The current manuscript is a computational study, so no data have been generated for this manuscript. All simulation code used for this paper is available on GitHub, copy archived at *Wilmes, 2025*.

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

# Appendix 1

## Synaptic dynamics/plasticity rules

$$\Delta w_{\text{PV}^+,\text{SST}^+} = \eta_{\text{PV}}(w_{\text{PV}^+,s}\text{s} - w_{\text{PV}^+,\text{SST}^+}r_{\text{SST}^+})r_{\text{SST}^+}, \tag{29}$$

$$\Delta w_{\text{PV}^-,\text{SST}^-} = \eta_{\text{PV}}(w_{\text{PV}^-,R}r_{\text{R}} - w_{\text{PV}^-,\text{SST}^-}r_{\text{SST}^-})r_{\text{SST}^-}, \tag{30}$$

$$\Delta w_{\text{UPE}^+,\text{SST}^+} = \eta_{\text{UPE}}(w_{\text{UPE}^+,s}\text{s} - w_{\text{UPE}^+,\text{SST}^+}r_{\text{SST}^+})r_{\text{SST}^+}, \tag{31}$$

$$\Delta w_{\text{UPE}^+\text{SST}^-} = \eta_{\text{UPE}}(w_{\text{UPE}^-},Rr_R - w_{\text{UPE}^-,\text{SST}} - r_{\text{SST}^-}), \tag{32}$$

## Hierarchical predictive coding with uncertainties

The total energy across the $w_{\text{PV}^{+/-},\text{a}}$ layers is

$$E = \frac{1}{2}\sum_{\ell=0}^{L-1}\|\frac{e_\ell}{\sigma_\ell}\|^2 , \tag{33}$$

where $\sigma_\ell$ and $e_\ell$ are vectors, and the division is taken component-wise. The error at layer $\ell$ is

$$e_\ell = r_\ell - \rho(W_\ell r_{\ell+1}) , \tag{34}$$

where, $r_0 = s$ is the stochastic sensory input vector, and $r_L$ represents a given context vector at the very top layer $L$. The only source of stochasticity is the stimulus $s$, for which the top-down input, $\rho(W_0 r_1)$, is an estimate of its mean.

### Dynamics for uncertainty-weighted prediction errors

We first consider the gradient dynamics $\dot{r}_\ell = -\frac{\partial E}{\partial r_\ell}$ for a fixed stimulus $s$, and assume that this converges to a critical point of $E$ given by $\frac{\partial E}{\partial r_\ell} = 0$. In our simulations, $s$ is sampled from a Gaussian with constant mean, or a mean that changes in steps across time. We calculate (assuming that the variances $\sigma_\ell^2$ are constant),

$$\dot{r}_\ell = -\frac{\partial E}{\partial r_\ell} = W_{\ell-1}^T\left(\rho'(W_{\ell-1}r_\ell)\cdot\frac{e_{\ell-1}}{\sigma_{\ell-1}^2}\right) - \frac{e_l}{\sigma_l^2} , \tag{35}$$

where $\cdot$ denotes the component-wise product. To reveal the relation to our uncertainty-weighted prediction errors UPEs, we rewrite this equation as

$$\dot{r}_\ell = M_\ell\frac{e_{\ell-1}}{\sigma_{\ell-1}^2} - \frac{e_l}{\sigma_l^2} = M_\ell\,\text{UPE}_{\ell-1} - \text{UPE}_\ell , \tag{36}$$

with matrix $M_\ell$ mapping from the lower layer $\ell - 1$ to layer $\ell$,

$$M_\ell = W_{\ell-1}^T\,\text{diag}\big(\rho'(W_{\ell-1}r_\ell)\big) . \tag{37}$$

*Equation 36* is the general rate dynamics in a hierarchical predictive network. It represents a reformulation of the classical predictive coding model by *Rao and Ballard, 1999*, but with a noise model that is restricted to Gaussian noise only in the stimulus $s$. Upper layer representations $r_\ell$ will inherit the noise from the sensory input by a propagation of the stochastic error $e_0 = s - \rho(W_0 r_1)$ to higher layers $\ell$. We also consider a fixed prior on the context rate $r_L$ at the top layer $L$ that is not included in the energy.

Going beyond (*Rao and Ballard, 1999*), we show how a microcircuit can explicitly learn the uncertainties $\sigma_\ell^2$, and how they can scale the error representations. Notice that the dynamics of *Equation 36* only contains divisions by $\sigma^2$ in the UPEs, and hence will lead to a simple neuronal implementation as shown in *Figure 8* for two layers.

From *Equation 36* we obtain the special case of *Equation 6* in the main text, that is $\dot{r} = (\text{UPE}_s^+ - \text{UPE}_s^-) - (\text{UPE}_p^+ - \text{UPE}_p^-)$, by choosing $\ell = 1$, $M_\ell$ the identity, and decomposing the uncertainty-modulated prediction error into the positive and negative part, $\text{UPE}_\ell = \text{UPE}_\ell^+ - \text{UPE}_\ell^-$.

Opposite of $r_\ell$ scaling by upper and lower uncertainties.

We next give two expressions for the steady state rate reached at $\dot{r}_\ell = 0$ with a given stimulus $s$. The steady state condition $\dot{r}_\ell = -\frac{\partial E}{\partial r_\ell} = 0$ writes according to **Equation 36** as

$$M_\ell \frac{e_{\ell-1}}{\sigma_{\ell-1}^2} - \frac{e_l}{\sigma_l^2} = 0 \,, \tag{38}$$

where $M_\ell$ is defined in **Equation 37**. To lighten the notation, we abbreviate

$$\hat{r}_\ell = \rho(W_\ell r_{\ell+1}) \,. \tag{39}$$

$\hat{r}_\ell$ is the top-down estimate of the representation $r_\ell$, based on the prior $r_{\ell+1}$ in the upper layer. The error defined in **Equation 34** then writes as $e_l = r_\ell - \hat{r}_\ell$. Plugging this into **Equation 38**, yields a self-consistency equation for $r_\ell$

$$r_\ell = \hat{r}_\ell + \sigma_\ell^2 \cdot \left( M_\ell \frac{e_{\ell-1}}{\sigma_{\ell-1}^2} \right) \,. \tag{40}$$

Here, the divisive modulation of the error by the lower-layer uncertainty $\sigma_{\ell-1}^2$, and the multiplicative modulation by the upper-layer uncertainty $\sigma_\ell^2$ becomes visible. This feature is shared by the full model of uncertainty-coding that considers independent Gaussian noise at each layer, and that takes the rate-dependency of $\sigma_\ell^2$ into account (resulting in second-order error driving the dynamics of $r_\ell$, see **Granier et al., 2024**). Crucially, again, the neuronal implementation of the dynamics in **Equation 36** only requires the division by $\sigma_{\ell-1}^2$ and $\sigma_\ell^2$, see also **Figure 8**.

## Uncertainty representation as a convex combinations of rates

According to the dynamics in **Equation 36**, the dynamics of the representation is given by a combination of bottom-up and top-down errors. The steady state is characterised by balanced errors, $M_\ell \mathrm{UPE}_{\ell-1} = \mathrm{UPE}_\ell$. We next show that also on the level of the rates, the steady state can be written as a combination of bottom-up and top-down rates.

For this, we introduce the bottom-up error-corrected representation $\check{r}_\ell$, which is the representation $r_\ell$ updated by the uncertainty-weighted error from the lower layer,

$$\check{r}_\ell = r_\ell + M_\ell \frac{e_{\ell-1}}{\sigma_{\ell-1}^2} \,. \tag{41}$$

From this we obtain $\sigma_\ell^2$, while $r_\ell - \hat{r}_\ell = e_l$ is the error from **Equation 34**. Plugging these two expressions into the dynamics underlying **Equation 38** yields

$$(\check{r}_\ell - r_\ell) + \frac{\hat{r}_\ell - r_\ell}{\sigma_l^2} = 0 \,. \tag{42}$$

This **Equation 42** now yields a self-consistency equation for $r_\ell$, where the representation $r_\ell$ is a convex combination of the top-down prior $\hat{r}_\ell$ and the bottom-up update $\check{r}_\ell$,

$$r_\ell = \frac{1}{c} \left( \frac{\hat{r}_\ell}{\sigma_l^2} + \check{r}_\ell \right) \,. \tag{43}$$

Here, $c = \frac{1}{\sigma_\ell^2} + \mathbf{1}$ is the sum of the certainty vector $1/\sigma_\ell^2$ at layer $\ell$ and the vector $1$ of dim $(r_\ell)$ filled by 1's. Hence, the integration of the UPEs according to **Equation 36** converges to the convex combination of $r_\ell$ and $\check{r}_\ell$ given by **Equation 43**. This is the generalization of **Equation 4** in the main text.

## Interpretation of the convex combination

**Equation 43** can be interpreted in different ways: It gives

- the posterior rate $r_\ell$ as convex combination of the top-down prior $\hat{r}_\ell$ (**Equation 43**) and the bottom-up error-corrected representation $\check{r}_\ell$ (**Equation 39**). Notice that 'prior' and 'posterior' here do not imply the classical Bayesian inversion since the noise at the various layers $\ell$, inherited from the stochastic stimulus $r_\ell$, is not independent.

- a self-consistency equation for the stationary rate $r_\ell$ satisfying $\frac{\partial E}{\partial r_\ell} = 0$, with $r_\ell$ appearing also on the right-hand side of *Equation 43* via *Equations 37 and 41*.
- an iterative scheme to calculate the steady-state representation $r_\ell$, starting from the old value of $r_\ell$ on the right-hand side, and obtaining the new value on the left-hand side, if the iteration converges. The iteration typically converges due to the gradient descent construction.

