## [Editor Report · eLife Assessment]

This **important** study introduces a new cortical circuit model for predictive processing. Simulations effectively illustrate that, with appropriate synaptic plasticity, a canonical layer 2/3 cortical circuit - comprising two classes of interneurons providing subtractive and divisive inhibition - can generate uncertainty-modulated prediction errors by pyramidal neurons. The model is **compelling**; although it relies on many assumptions and has not yet been compared directly to data, the model does align with empirical observations and yields a range of testable predictions. The study is expected to be of great interest to those involved in cortical and predictive processing research.

---

## [Referee Report · Reviewer #2 (Public review)]

Summary:

This computational modeling study addresses the observation that variable observations are interpreted differently depending on how much uncertainty an agent expects from its environment. That is, the same mismatch between a stimulus and an expected stimulus would be less significant, and specifically would represent a smaller prediction error, in an environment with a high degree of variability than in one where observations have historically been similar to each other. The authors show that if two different classes of inhibitory interneurons, the PV and SST cells, (1) encode different aspects of a stimulus distribution and (2) act in different (divisive vs. subtractive) ways, and if (3) synaptic weights evolve in a way that causes the impact of certain inputs to balance the firing rates of the targets of those inputs, then pyramidal neurons in layer 2/3 of canonical cortical circuits can indeed encode uncertainty-modulated prediction errors. To achieve this result, SST neurons learn to represent the mean of a stimulus distribution and PV neurons its variance.

The impact of uncertainty on prediction errors in an understudied topic, and this study provides an intriguing and elegant new framework for how this impact could be achieved and what effects it could produce. The ideas here differ from past proposals about how neuronal firing represents uncertainty. The developed theory is accompanied by several predictions for future experimental testing, including the existence of different forms of coding by different subclasses of PV interneurons, which target different sets of SST interneurons (as well as pyramidal cells). The authors are able to point to some experimental observations that are at least consistent with their computational results. The simulations shown demonstrate that if we accept its assumptions, then the authors' theory works very well: SSTs learn to represent the mean of a stimulus distribution, PVs learn to estimate its variance, firing rates of other model neurons scale as they should, and the level of uncertainty automatically tunes the learning rate, so that variable observations are less impactful in a high uncertainty setting.

Strengths:

The ideas in this work are novel and elegant, and they are instantiated in a progression of simulations that demonstrate the behavior of the circuit. The framework used by the authors is biologically plausible and matches some known biological data. The results attained, as well as the assumptions that go into the theory, provide several predictions for future experimental testing. The authors have taken into account earlier review comments to revise their paper in ways that enhance its clarity.

Weaknesses:

One weakness could be that the proposed theory does rely on a fairly large number of assumptions. However, there is at least some biological support for these. Importantly, the authors do lay out and discuss their key assumptions in the Discussion section, so readers can assess their validity and implications for themselves.

Comments on revisions:

I have no further suggestions for the authors.

---

## [Referee Report · Reviewer #4 (Public review)]

Summary:

Wilmes and colleagues develop a model for the computation of uncertainty modulated prediction errors based on an experimentally inspired cortical circuit model for predictive processing. Predictive processing is a promising theory of cortical function. An essential aspect of the model is the idea of precision weighting of prediction errors. There is ample experimental evidence for prediction error responses in cortex. However, a central prediction of the theory is that these prediction error responses are regulated by the uncertainty of the input. Testing this idea experimentally has been difficult due to a lack of concrete models. This work provides one such model and makes experimentally testable predictions.

Strengths:

The model proposed is novel and well-implemented. It has sufficient biological accuracy to make useful and testable predictions.

Weaknesses:

One key idea the model hinges on is that stimulus uncertainty is encoded in the firing rate of parvalbumin positive interneurons. While this assumption is rather speculative, the model also here makes experimentally testable predictions.

Comments on revisions:

Congratulations on a very nice paper.

---

## [Author Response]

The following is the authors’ response to the previous reviews.

**Public Reviews:**

**Reviewer #2 (Public Review):**
Summary:This computational modeling study addresses the observation that variable observations are interpreted differently depending on how much uncertainty an agent expects from its environment. That is, the same mismatch between a stimulus and an expected stimulus would be less significant, and specifically would represent a smaller prediction error, in an environment with a high degree of variability than in one where observations have historically been similar to each other. The authors show that if two different classes of inhibitory interneurons, the PV and SST cells, (1) encode different aspects of a stimulus distribution and (2) act in different (divisive vs. subtractive) ways, and if (3) synaptic weights evolve in a way that causes the impact of certain inputs to balance the firing rates of the targets of those inputs, then pyramidal neurons in layer 2/3 of canonical cortical circuits can indeed encode uncertainty-modulated prediction errors. To achieve this result, SST neurons learn to represent the mean of a stimulus distribution and PV neurons its variance.The impact of uncertainty on prediction errors in an understudied topic, and this study provides an intriguing and elegant new framework for how this impact could be achieved and what effects it could produce. The ideas here differ from past proposals about how neuronal firing represents uncertainty. The developed theory is accompanied by several predictions for future experimental testing, including the existence of different forms of coding by different subclasses of PV interneurons, which target different sets of SST interneurons (as well as pyramidal cells). The authors are able to point to some experimental observations that are at least consistent with their computational results. The simulations shown demonstrate that if we accept its assumptions, then the authors’ theory works very well: SSTs learn to represent the mean of a stimulus distribution, PVs learn to estimate its variance, firing rates of other model neurons scale as they should, and the level of uncertainty automatically tunes the learning rate, so that variable observations are less impactful in a high uncertainty setting.Strengths:The ideas in this work are novel and elegant, and they are instantiated in a progression of simulations that demonstrate the behavior of the circuit. The framework used by the authors is biologically plausible and matches some known biological data. The results attained, as well as the assumptions that go into the theory, provide several predictions for future experimental testing. The authors have taken into account earlier review comments to revise their paper in ways that enhance its clarity.Weaknesses:One weakness could be that the proposed theory does rely on a fairly large number of assumptions. However, there is at least some biological support for these. Importantly, the authors do lay out and discuss their key assumptions in the Discussion section, so readers can assess their validity and implications for themselves.

Thank you very much, we are very satisfied with this public review.

**Reviewer #4 (Public Review):**
Summary:Wilmes and colleagues develop a model for the computation of uncertainty modulated prediction errors based on an experimentally inspired cortical circuit model for predictive processing. Predictive processing is a promising theory of cortical function. An essential aspect of the model is the idea of precision weighting of prediction errors. There is ample experimental evidence for prediction error responses in cortex. However, a central prediction of the theory is that these prediction error responses are regulated by the uncertainty of the input. Testing this idea experimentally has been difficult due to a lack of concrete models. This work provides one such model and makes experimentally testable predictions.Strengths:The model proposed is novel and well-implemented. It has sufficient biological accuracy to make useful and testable predictions.Weaknesses:One key idea the model hinges on is that stimulus uncertainty is encoded in the firing rate of parvalbumin positive interneurons. This assumption, however, is rather speculative and there is no direct evidence for this.

Thank you very much for this nice description. With regard to the weakness: it is true that the key idea hinges on uncertainty being encoded in the firing of inhibitory neurons. If it turns out that these inhibitory neurons are not PV neurons, however, the theory does not break down. The suggestion of PV neurons is fueled by the observation that PV neurons implement shunting and hence divisive inhibition and by the connectivity of PVs in the circuit. We discuss this in the discussion section: "To provide experimental predictions that are immediately testable, we suggested specific roles for SSTs and PVs, as they can subtractively and divisively modulate pyramidal cell activity, respectively. In principle, our theory more generally posits that any subtractive or divisive inhibition could implement the suggested computations. With the emerging data on inhibitory cell types, subtypes of SSTs and PVs or other cell types may turn out to play the proposed role."

**Recommendations for the authors:**

**Reviewer #4 (Recommendations For The Authors):**
(1) Line numbers would simplify reviewing.

We will add line numbers to our next submission.

(2) The existence of positive and negative PE was already suggested by Rao & Ballard.

We added the citation to the sentence "Because baseline firing rates are low in layer 2/3 pyramidal cells () positive and negative prediction errors were suggested to be represented by distinct neuronal populations [44,66],[...]" in the section "Computation of UPEs in cortical microcircuits".

(3) wekk should probably read well.

Indeed, thank you. We fixed it.

(4) Figure 4. legends A-C are mixed up. What are the two values of ¦s-u¦ in F and I - the same as in D and F.

Thank you, we fixed this.

(5) "representation neurons, the activity of which reflects the internal model". For consistency with the original definitions this should read "the activity of which reflects the internal representation". The internal "model" is the synaptic weights (or transformation between areas) - the activity of representation neurons (as the name implies) is the internal "representation".

Thank you, we changed it.

(6) "Mice trained in a predictable environment [...] [4]." This should read "reared" in an unpredictable environment, etc. Relatedly, the problem with this argument is that, the referenced paper argues that the mice never learned to predict and the reduced PE responses are a consequence of a reduction in prediction strength (these mice never - in life - had experience of visuomotor coupling). Better evidence might be the acute changes observed in normal mice (see e.g. Figure 3B in https://pubmed.ncbi.nlm.nih.gov/22681686/). However, another finding from the paper referenced is that in mice reared without visuomotor coupling, MM responses of SST interneurons are unchanged, while those in PV interneurons are completely absent. Would the authors model come to similar results if trained in an environment with (very) high uncertainty and then tested in a low uncertainty environment?

Thank you for pointing us to Figure 3B of Keller et al. 2012. We are now citing this result as it is indeed better evidence.

Thank you very much for your illuminating question and for pointing out that a mouse that never experienced a predictable visual flow may not have formed a model of the visual flow, and hence may not have any prediction about its visual experience. We haven’t considered this scenario in our paper before. So far, we only considered scenarios, in which it is possible to learn a prediction, i.e. to infer the mean from the sensory input. We now consider this other scenario in which the mouse that was reared in an unpredictable environment did not form a prediction and compare SST (1) and PV (2) activity in this mouse to one that learned to form a prediction, and added it to the section "Predictions for different cell types":

"Second, prediction error activity seems to decrease in less predictable, and hence more uncertain, contexts: in mice reared in a predictable environment [where locomotion and visual flow match, 42], error neuron responses to mismatches in locomotion and visual flow decreased with each day of experiencing these unpredictable mismatches. Third, the responses of SSTs and PVs to mismatches between locomotion and visual flow [4] are in line with our model (note that in this experiment the mismatches are negative prediction errors as visual flow was halted despite ongoing locomotion): In this study, SST responses decreased during mismatch, i.e. when the visual flow was halted, and there was no difference between mice reared in a predictable or unpredictable environment. In line with these observations, the authors concluded that SST responses reflected the actual visual input. In our model negative PE circuit, SSTs also reflect the actual stimulus input, which in our case was a whisker stimulus (SST rates in Fig. 6C and I reflect the stimuli (black and grey bar) in A and G, respectively) and SST rates are the same for high and low uncertainty (corresponding to mice reared in a predictable or unpredictable environment). In the same study, PV responses were absent towards mismatches in animals reared in an unpredictable environment [4]. The authors argued that mice reared in an unpredictable environment did not learn to form a prediction. In our model, the missing prediction corresponds to missing predictive input from the auditory domain (e.g. due to undeveloped synapses from the predictive auditory input). If we removed the predictive input in our model, PVs in the negative PE circuit would also be silent as they would not receive any of the excitatory predictive inputs."

(7) "Our model further posits the existence of two distinct subtypes of SSTs in positive and negative error circuits." There is some evidence for this: Figure 5a in https://pubmed.ncbi.nlm.nih.gov/36747710/

Thank you, we added this citation to the corresponding section.